# mTORC1 regulates cell survival under glucose starvation through 4EBP1/2-mediated translational reprogramming of fatty acid metabolism

Tal Levy[1,2,21], Kai Voeltzke[3,21], Laura Hruby[3,21], Khawla Alasad [1,2,21], Zuelal Bas[3], Marteinn Snaebjörnsson [4,5], Ran Marciano[1,2], Katerina Scharov[3,6], Mélanie Planque [7,8], Kim Vriens[7,8], Stefan Christen[7,8], Cornelius M. Funk [9,10], Christina Hassiepen[3], Alisa Kahler [3], Beate Heider[3], Daniel Picard [3,6,11], Jonathan K. M. Lim [3], Anja Stefanski[12], Katja Bendrin[13], Andres Vargas-Toscano [14,15,16], Ulf D. Kahlert [17], Kai Stühler[12], Marc Remke [3,6,11], Moshe Elkabets[18,19], Thomas G. P. Grünewald [9,10,20], Andreas S. Reichert [13], Sarah-Maria Fendt [7,8], Almut Schulze [4,5], Guido Reifenberger[3,11], Barak Rotblat [1,2,22] ✉ & Gabriel Leprivier [3,22] ✉

Energetic stress compels cells to evolve adaptive mechanisms to adjust their metabolism. Inhibition of mTOR kinase complex 1 (mTORC1) is essential for cell survival during glucose starvation. How mTORC1 controls cell viability during glucose starvation is not well understood. Here we show that the mTORC1 effectors eukaryotic initiation factor 4E binding proteins 1/2 (4EBP1/2) confer protection to mammalian cells and budding yeast under glucose starvation. Mechanistically, 4EBP1/2 promote NADPH homeostasis by preventing NADPH-consuming fatty acid synthesis via translational repression of Acetyl-CoA Carboxylase 1 (ACC1), thereby mitigating oxidative stress. This has important relevance for cancer, as oncogene-transformed cells and glioma cells exploit the 4EBP1/2 regulation of ACC1 expression and redox balance to combat energetic stress, thereby supporting transformation and tumorigenicity in vitro and in vivo. Clinically, high *EIF4EBP1* expression is associated with poor outcomes in several cancer types. Our data reveal that the mTORC1-4EBP1/2 axis provokes a metabolic switch essential for survival during glucose starvation which is exploited by transformed and tumor cells.

Glucose is one of the most important nutrients, and its shortage in living organisms represents a cardinal physiological stress. Indeed, glucose deprivation compels cells to evolve molecular mechanisms to adjust their metabolism and sustain survival. These mechanisms are particularly critical for cells residing within solid tumors, as these tissues suffer from glucose deprivation due to defective vascularization[1–3]. Metabolic adaptation to glucose starvation occurs through the blocking of anabolic processes, paralleled with the activation of catabolic processes, which together preserve cellular energy and redox balance[4]. These responses are coordinated by highly conserved energy sensors and signaling hubs, such as AMP-activated protein kinase (AMPK)[5,6] and mechanistic target of rapamycin complex

1 (mTORC1)[7,8], both essential for determining cell fate under glucose-deprived conditions.

AMPK is activated by glucose deprivation following direct sensing of increases in AMP/ATP and ADP/ATP ratios, as well as depletion of fructose-1,6-bisphosphate[5,6]. Upon activation, AMPK phosphorylates several well-defined substrates to increase cellular energy production and reduce energy consumption, as well as to reprogram metabolism[5,6]. Specifically, during glucose starvation, AMPK phosphorylates and inhibits Acetyl-CoA carboxylase 1 (ACC1), the rate-limiting enzyme of the fatty acid synthesis pathway, to reduce fatty acid synthesis[9]. Since this metabolic process is the most NADPH-consuming process in a cell, AMPK-mediated blockage of ACC1 enables the maintenance of NADPH homeostasis, curbing reactive oxygen species (ROS), and promoting cell survival in glucose-deprived conditions[9].

Mirroring AMPK, mTORC1 is inactivated by glucose deprivation, which occurs through several distinct mechanisms[10]. These include AMPK-mediated phosphorylation of the mTORC1 component Raptor[11] and of the mTORC1 inhibitor TSC2[12], ULK1 (autophagy regulator)-mediated phosphorylation of the mTORC1 activator LARS1[13], as well as PFKFB3 and PFK1 (glycolytic enzymes)-mediated regulation of the mTORC1 activators RagA/B[14]. Given that mTORC1 controls protein, lipid, and nucleotide syntheses as well as autophagy[7,8], such inactivation leads to profound metabolic changes. While it is well established that mTORC1 inhibition is essential to protect cells suffering from glucose deprivation[12,15], the underlying mechanisms are poorly understood. In particular, it is not known which mTORC1 substrates and which metabolic processes are critical for mTORC1-mediated regulation of cell fate upon glucose deprivation.

Here we found that downstream of mTORC1, the mTORC1 substrates 4EBP1/2 act as evolutionarily conserved pro-survival factors during glucose deprivation. They function by reprogramming fatty acid metabolism to preserve NADPH homeostasis. 4EBP1/2 are known inhibitors of mRNA translation initiation induced by metabolic stress not limited to glucose starvation[16,17], that act by binding and restricting eukaryotic initiation factor 4E (eIF4E) to prevent cap-dependent mRNA translation initiation[18,19]. Specifically, we uncovered that during glucose starvation, 4EBP1/2 protect cells by translationally reducing the expression of ACC1 to block fatty acid synthesis and maintain redox homeostasis. While the role of 4EBP1/2 in cancer is still under debate[20], our findings reveal that 4EBP1 promotes tumorigenesis of oncogene-transformed and glioma cells by restricting ACC1 expression. Moreover, we showed that high expression of *EIF4EBP1* (the gene encoding 4EBP1) is a factor of poor prognosis in glioma and glioblastoma. These findings unravel the biological and pathological basis of mTORC1 function in energetically challenged cells and bring forth 4EBP1/2 as previously undescribed fatty acid synthesis inhibitors.

## Results

### mTORC1 inhibition protects against glucose starvation through 4EBP1/2 mediated eIF4E inhibition

To investigate the contribution of mTORC1 inhibition to survival during glucose starvation, and to identify the contributing mTORC1 effectors, we used AMPK knockout (KO) mouse embryonic fibroblasts (MEFs) and HeLa cells. These cells are highly sensitive to glucose starvation due to AMPK absence or dysregulation, respectively[9,21]. Indeed, AMPK KO and HeLa cells exhibited significant levels of cell death when subjected to glucose starvation, and treatment with the mTORC1 inhibitor rapamycin did not rescue such cell death (Fig. 1A–C), as previously reported[9]. Surprisingly, treatment with a dual mTORC1/mTORC2 inhibitor, Ku-0063794 (KU), significantly reduced the rates of cell death upon glucose starvation in both cell lines (Fig. 1A–C). One ostensible difference between these mTOR inhibitors is that in contrast to KU, rapamycin is inefficient at blocking mTORC1-mediated phosphorylation of its substrates 4EBP1/2[22]. We reasoned that since mTORC1-mediated 4EBP1/2 phosphorylation is inhibitory,

KU, but not rapamycin, may be triggering 4EBP1/2 activation, pointing to a possible role of 4EBP1/2 in the mTORC1-mediated regulation of cell fate under glucose restriction (Fig. 1C). To test our theory, we used an eIF4E inhibitor, 4EGI[23], to mimic the action of 4EBP1/2, or a protein synthesis inhibitor, cycloheximide (CHX), since the activation and subsequent binding of 4EBP1/2 to eIF4E inhibits the formation of the cap-dependent mRNA translation initiation complex. Indeed, we discovered that 4EGI or CHX treatment markedly protected cells exposed to glucose deprivation as efficiently as KU (Fig. 1A-C). In further support of the role of 4EBP1/2 in protecting cells against glucose starvation, overexpression of a constitutively active 4EBP1 mutant 4EBP1 (T37A/T46A) (4EBP1^AA) was sufficient to fully protect AMPK-dysregulated HeLa cells from the induction of cell death under glucose starvation (Fig. 1D), while an eIF4E-non-binding mutant of 4EBP1^AA, 4EBP1 (Y54A/L59A) (4EBP1^AA, YL [24]), failed to prevent cell death in these conditions (Fig. 1E). Conversely, double knockout of 4EBP1/2 (4E KO) rendered mouse embryonic fibroblasts (MEFs) highly sensitive to glucose starvation, as seen with the high rates of cell death compared to 4EBP1/2 wild type (WT) MEFs (Fig. 1F). Ectopic expression of 4EBP1^AA was able to rescue 4E KO MEFs from glucose starvation-induced cell death (Fig. S1A).

Following our findings above, we wondered whether 4EBP1/2 could exert a general pro-survival function under glucose deprivation in other cell lines. Indeed, targeting the expression of 4EBP1/2 or 4EBP1 alone, by stable knockdown (kd), severely impaired the survival of numerous cell lines under glucose deprivation, but not under basal conditions (Fig. 1F, G, Fig. S1B–I). These included cells of diverse lineages such as human embryonic kidney (HEK) 293 cells, induced pluripotent stem cells (iPSC), breast cancer cells (MCF7), neuroblastoma cells (IMR-32 and Kelly), and medulloblastoma cells (HD-MB03 and Med8a) (Fig. 1G, Fig. S1B–I). Furthermore, 4EBP1/2 depletion was sufficient to sensitize MEFs and HEK293 cells to low glucose conditions (Fig. S1J, K).

We sought to further corroborate the involvement of eIF4E in the protective function of 4EBP1/2. Notably, kd of eIF4E in either 4E KO MEFs or 4EBP1/2 kd (sh4EBP1/2) HEK293 cells led to a significant reduction of cell death under glucose starvation (Fig. 1H, Fig. S1L), which is in line with data obtained with the 4EBP1^AA, YL mutant (Fig. 1E). Altogether, these strongly indicate that 4EBP1/2 protect cells from glucose starvation by binding to and inhibiting eIF4E.

Since the mTORC1 pathway is highly conserved, we also asked whether the 4EBP1/2 protective function represents an evolutionarily conserved biological response to glucose deprivation. To answer this we employed the use of *Saccharomyces cerevisiae* as a model organism, since yeast possess two functional 4EBP orthologues, Eap1p and Caf20p[25,26]. While disruption of *eap1* (*eap1Δ*) or *caf20* (*caf20Δ*) had either a small or no observable impact, respectively, on the growth capacities in basal, glucose-containing YPD medium, the growth of *eap1Δ* strain, but not of *caf20Δ* strain, was severely compromised in glucose-free YP media, as compared to WT strain (Fig. 1I). Deletion of both *eap1* and *caf20* (*eap1Δ/caf20Δ*) had no additional detrimental impact on growth in glucose-free medium compared to *eap1Δ* strain alone (Fig. 1I). Moreover, clonogenic assays of WT and *eap1Δ* strains grown in liquid conditions in the absence of glucose indicated that disruption of *eap1* prevented the survival of yeast upon glucose withdrawal (Fig. 1J), further supporting that the yeast 4EBP orthologue Eap1p mediates pro-survival properties in glucose deprived conditions.

Together, these data provide evidence that 4EBP1/2 are conserved pro-survival factors in the response to glucose deprivation that act by binding and inhibiting eIF4E.

### 4EBP1/2 maintain NADPH homeostasis and the redox balance under glucose deprivation

Next, we sought to understand the cellular processes that mediate the 4EBP1/2 protective function under glucose starvation. Since it is well

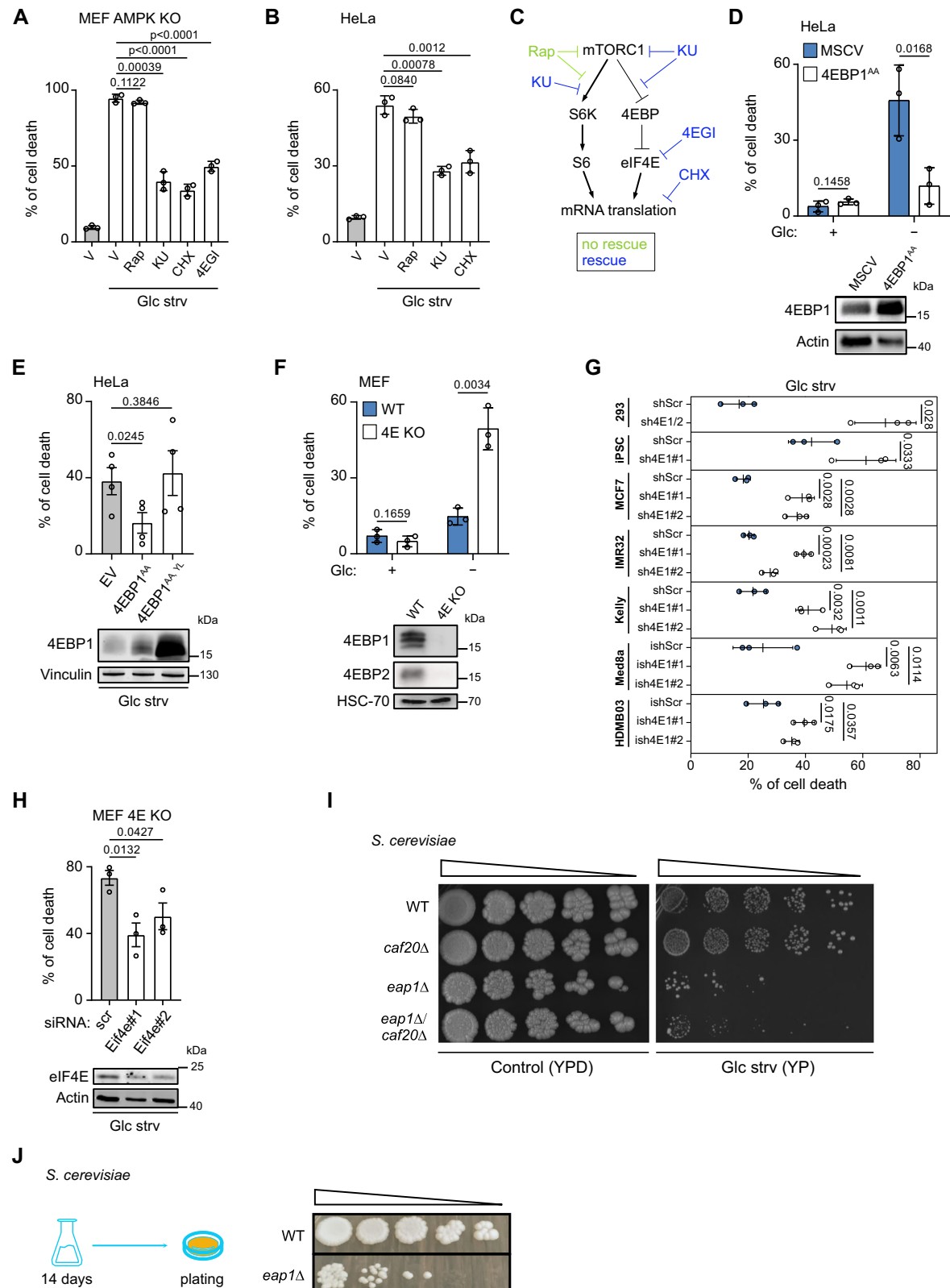

established that 4EBP1/2 repress overall protein synthesis[16,17], proliferation, and mitochondrial activity[27,28], we first assessed the impact of 4EBP1/2 on these cellular processes in response to glucose deprivation. To our surprise, 4EBP1/2 depletion in MEFs and HEK293 cells did not impact the rates of overall protein synthesis under glucose starvation (Fig. 2A, Fig. S2A), as determined using azidohomoalanine (AHA) labeling and Click Chemistry[29]. In contrast, 4EBP1/2 were

essential for the reduction of global protein synthesis in normal media upon treatment with the mTORC1 inhibitor KU (Fig. S2B), consistent with previous reports[27]. We reasoned that since 4EBP1/2 protection during glucose starvation is dependent upon eIF4E binding, and since inhibition of protein synthesis through cycloheximide treatment rescued HeLa and AMPK KO cells from glucose starvation, 4EBP1/2 may protect cells by reducing the synthesis of specific proteins rather than

**Fig. 1 | The mTORC1 substrates 4EBP1/2 prevent cell death in response to glucose starvation in human, mouse and yeast cells. A, B** The indicated cell lines were grown in glucose starved medium (Glc strv) and treated with vehicle (V), Rapamycin (Rap), KU-0063794 (KU), cycloheximide (CHX) or 4EGI for 48 h. Cell death was measured by PI staining and flow cytometry. **C** Scheme of the mTORC1 downstream signaling pathways controlling mRNA translation initiation and the impact of the inhibitors used in (**A, B**). **D** The indicated cell lines were grown in complete media or starved for glucose (Glc) for 48 h. Cell death was analyzed as in (**A, B**). The level of the indicated proteins was analyzed by immunoblotting. **E** The indicated cell lines were grown in glucose starved medium (Glc strv) for 48 h. Cell death and protein levels were analyzed as in (**A, B**) and (**D**) respectively. **F** The indicated cell lines were grown in complete media or starved for glucose (Glc) for 48 h. Cell death and protein levels were analyzed as in (**A, B**) and (**C**) respectively.

**G** The indicated cell lines were glucose (Glc) starved for 48 h. Med8a and HD-MB03 cells were treated with 1 µg/ml doxycycline for 72 h. Cell death was measured as in (**A, B**). **H** 4E KO MEF were transfected with control siRNA (scr) or siRNAs targeting *Eif4e* and grown in glucose starved medium (Glc strv) for 48 h. Cell death and protein levels were analyzed as in (**A, B**) and (**D**) respectively. **I** WT, *caf20Δ*, *eap1Δ*, or *eap1Δ/caf20Δ S. cerevisiae* strains were plated by serial dilution on solid complex medium with (YPD) or without (YP) 2% glucose at 37 °C. **J** WT or *eap1Δ* strains were grown in liquid medium containing no glucose (YP) for 2 weeks at 30 °C and were plated by serial dilutions onto complete YPD agar plates. Data are shown as the mean ± SD. Statistics: unpaired one-sided Student's *t* test (**A, B, D, E, F, G, H**); *n* = 3 independent experiments for (**A, B, D, E, F, G, H**). Source data are provided as a Source Data file.

inhibiting total cap-dependent mRNA translation. We also found no observable impact of 4EBP1/2 on proliferation, as proliferation rates were severely reduced following 24 h glucose deprivation in all cell lines tested, irrespective of their 4EBP1/2 status (Fig. 2B, Fig. S2C). Similarly, expression of 4EBP1/2 had no consistent impact on the relative mitochondrial membrane potential under glucose starvation, in these same cell lines (Fig. 2C, Fig. S2D), ruling out the inhibition of proliferation or mitochondrial regulation by 4EBP1/2 as explaining factors for their pro-survival functions in glucose starved cells. Additionally, we ascertained that autophagy was not responsible for the observed function of 4EBP1/2 under glucose starvation, as rates of autophagy were similar in control and 4EBP1/2 deficient MEFs and HEK293 cells in such conditions (Fig. S2E, F). Importantly, loss of 4EBP1/2 did not preclude AMPK activation following glucose deprivation (Fig. S2G, H), suggesting that AMPK activation may be necessary but not sufficient for cellular protection under these conditions, and that 4EBP1/2 are additional, essential factors. Together, these data indicate that 4EBP1/2 protective function under glucose starvation is independent from regulation of global protein synthesis, cellular proliferation, mitochondrial activity, autophagy or AMPK activity.

To decipher the metabolic parameters that are controlled by 4EBP1/2 to promote survival under glucose starvation, we assessed the abundance of several metabolites involved in glucose metabolism in control and sh4EBP1/2 HEK293 cells upon 24 h glucose starvation. While we observed severe reduction in the levels of glycolytic and TCA cycle intermediates following glucose starvation in all cells, there was no major difference between control and 4EBP1/2 deficient HEK293 cells (Fig. 2D). In concordance, ATP levels were similar in control and 4EBP1/2 deficient HEK293 cells under glucose deprivation. This was unexpected as the regulation of cell fate by mTORC1 upon glucose withdrawal has been linked to preventing ATP consumption[15]. Instead, the strongest differences between control and 4EBP1/2 deficient HEK293 cells under glucose starvation were of NADH and NADPH levels, which were both decreased upon 4EBP1/2 loss (Fig. 2D). However, cellular NADH/NAD$^+$ ratio, which corresponds to the relative proportion of reduced to oxidized forms of this cofactor, was unchanged between control and sh4EBP1/2 HEK293 cells under glucose deprivation (Fig. 2E). Furthermore, NAD$^+$ and NADH levels were increased at 6 h glucose deprivation, but not at 1 h, with no consistent differences between 4EBP1/2 deficient and control HEK293 cells and MEFs (Fig. S2I–L). Strikingly, NADPH/NADP$^+$ ratio was more severely decreased in both 4EBP1/2 deficient MEFs and HEK293 cells compared to corresponding controls in glucose starved conditions (Fig. 2F, G). Conversely, 4EBP1$^{AA}$ overexpression led to a significant increase in NADPH/NADP$^+$ ratio in HeLa cells during glucose starvation (Fig. S3A), altogether suggesting that 4EBP1/2 augment NADPH levels under glucose depletion.

NADPH is an important cofactor for antioxidant reactions, in particular for the glutathione system, as it supports the recycling of oxidized glutathione GSSG to its reduced form GSH which is used to detoxify H$_2$O$_2$ (Fig. 2H). Thus, we investigated the impact 4EBP1/2 may

have on redox balance by measuring GSH/GSSG ratio and H$_2$O$_2$ levels. We observed that 4EBP1/2 deficient MEFs and HEK293 cells exhibited lower GSH/GSSG ratios, relative to their respective control cells under glucose starvation (Fig. 2I, Fig. S3B). In line with this observation, 4EBP1$^{AA}$ overexpression precluded severe depletion of GSH/GSSG ratio in HeLa cells upon glucose removal (Fig. S3C). Since total glutathione levels were unchanged in control and 4EBP1/2 deficient MEFs and HEK293, or in 4EBP1$^{AA}$ overexpressing cells during glucose deprivation (Fig. 2J, Fig. S3D, E), we surmised that 4EBP1/2 may be regulating glutathione recycling by maintaining NADPH levels, rather than glutathione biosynthesis. Accordingly, endogenous H$_2$O$_2$ levels were higher under glucose deprivation in 4EBP1/2 deficient MEFs and HEK293 cells compared to their respective controls (WT and shScr, respectively) (Fig. 2K, Fig. S3F). In addition, overexpression of 4EBP1$^{AA}$ prevented such increases in H$_2$O$_2$ levels in HeLa cells during glucose depletion (Fig. S3G). Finally, we sought to delineate the importance of 4EBP1/2 in regulating cellular redox balance towards cell survival upon glucose starvation. Notably, supplementation of glucose-starved 4E KO MEFs and sh4EBP1/2 HEK293 cells with antioxidants, including N-acetyl cysteine (NAC) or Catalase (CAT), significantly reduced cell death compared to vehicle-treated cells (Fig. 2L, Fig. S3H). Hence, our findings present strong evidence that 4EBP1/2 protect cells against glucose deprivation by promoting NADPH levels to mitigate oxidative stress.

## 4EBP1/2 control fatty acid synthesis under glucose deprivation to preserve NADPH levels

Following our above findings, we asked which metabolic processes 4EBP1/2 control to preserve NADPH levels and support cell survival upon glucose starvation. To answer this question, we used a proteomics approach to identify enzymes within NADPH producing or consuming pathways whose expression is affected by 4EBP1/2 under glucose deprivation. Proteome analysis comparing control and sh4EBP1/2 HEK293 cells under glucose starvation revealed differential expression of 9 enzymes from NADPH producing and 2 enzymes from NADPH consuming processes (Fig. 3A). In particular, we found these to include enzymes from three of the most NADPH producing processes – G6PD from oxidative pentose phosphate pathway, PHGDH, SHMT1, SHMT2, MTHFD1L and MTR from folate-mediated one-carbon pathway, and ME1, malic enzyme[30–32] – and enzymes of two of the most NADPH consuming processes, ACC1 from fatty acid synthesis and RRM1 from DNA synthesis[30] (Fig. 3A). We focused on the enzymes from NADPH producing processes that were downregulated in 4EBP1/2 deficient cells, as well as on enzymes from NADPH consuming processes that were increased, as this could explain the depletion in NADPH/NADP$^+$ ratio observed in these cells (Fig. 2F). We firstly excluded the involvement of malic enzyme, as ME1 was higher in sh4EBP1/2 versus control HEK293 cells. We also excluded folate-mediated one-carbon pathway, since the expression of some of these enzymes, PHGDH, SHMT2 and MTR, were lower in 4EBP1/2 deficient cells, while others, such as SHMT2 and MTHFD1L, were increased in the same cells

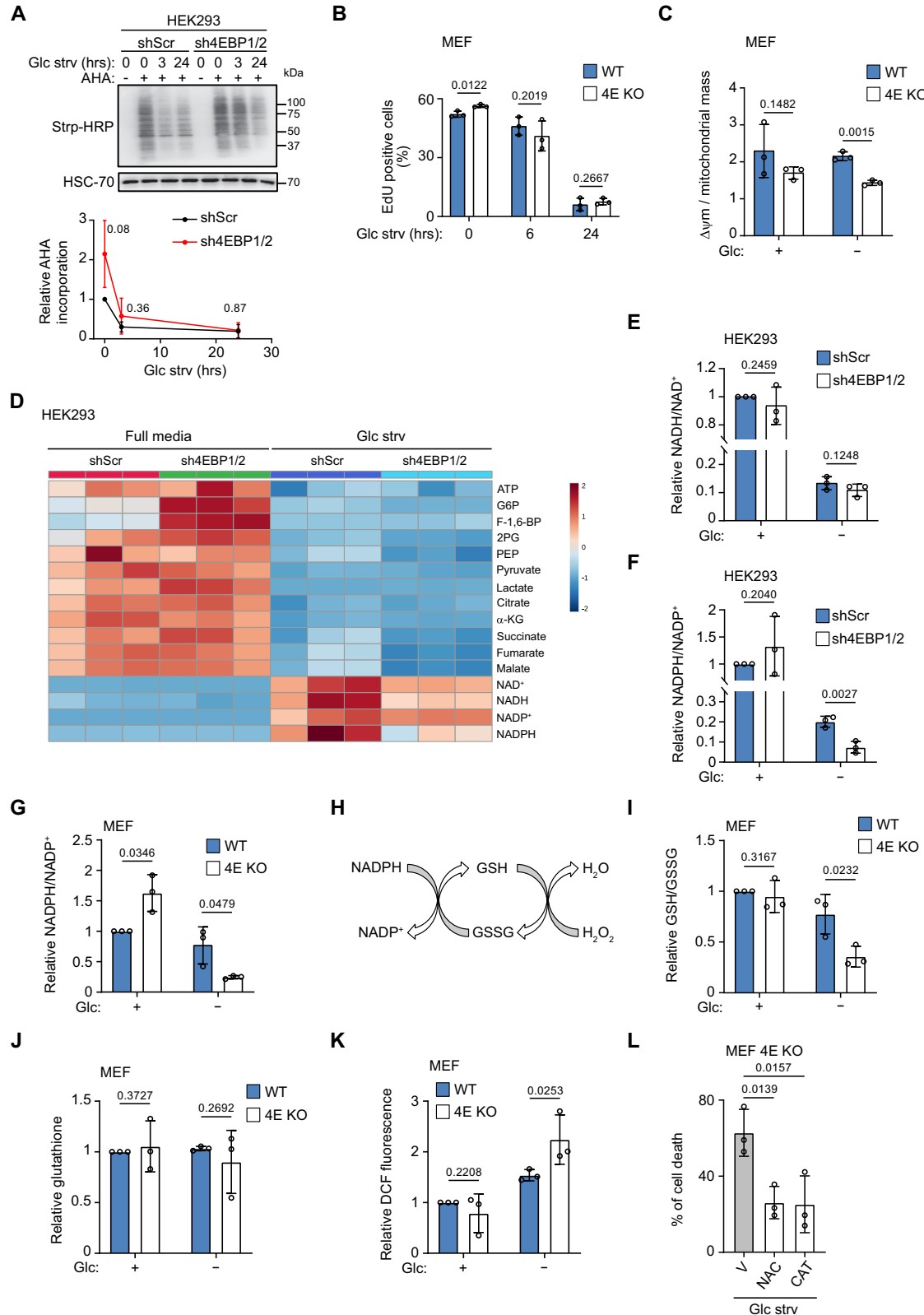

(Fig. 3A). IDH1 and IDH2, which contribute to NADPH production[32], were both lower in sh4EBP1/2 HEK293 cells (Fig. 3A). However, given that levels of α-ketoglutarate, the product of these enzymes, were not different between control and 4EBP1/2 deficient cells under glucose starvation (Fig. 2D), IDH1 and IDH2 were not further considered. From these analyses, G6PD, ACC1 and RRM1 remained as possible factors, thus pointing to oxidative pentose phosphate pathway, fatty acid

synthesis and DNA synthesis as potential metabolic processes controlled by 4EBP1/2 to maintain NADPH levels. However, inhibition of RRM1 with gemcitabine did not rescue 4E KO MEFs from glucose starvation-induced cell death (Fig. S4A), supporting that 4EBP1/2 protective functions are not related to increased RRM1 levels.

Since fatty acid synthesis is the most NADPH consuming process in a cell[30] and that its inhibition through regulation of ACC1 activity is

**Fig. 2 | 4EBP1/2 maintain antioxidant power and preserve the redox balance under glucose starvation. A** Control (shScr) and sh4EBP1/2 HEK293 cells were grown in complete medium or glucose starved (Glc strv) for the indicated times and labeled with azidohomoalanine (AHA). Levels of AHA-labelled proteins were detected by immunoblotting. **B** WT and 4E KO MEF were grown in complete medium or glucose starved (Glc strv) for the indicated times, labeled with EdU and analyzed by flow cytometry. **C** WT and 4E KO MEF were grown in complete medium or glucose (Glc) starved for 24 h. Mitochondrial membrane potential (Δψm) with TMRE staining and mitochondrial mass were measured. **D** Control (shScr) and sh4EBP1/2 HEK293 cells were grown in complete medium or glucose (Glc) starved for 24 h and the corresponding metabolites were measured by LC-MS. **E** NAD$^+$ and NADH levels measured in (**C**) were plotted as NADH/NAD$^+$ ratio. **F** NADP$^+$ and NADPH levels measured in (**C**) were plotted as NADPH/NADP$^+$ ratio. **G** WT and 4E KO MEF were grown in complete medium or glucose (Glc) starved for 24 h, and NADP$^+$ and NADPH levels were measured. **H** Scheme of the usage of NADPH in recycling oxidized glutathione for H$_2$O$_2$ detoxification. (**I**) WT and 4E KO MEF were grown as in (**C**), and reduced and total glutathione were measured and expressed as the ratio of reduced (GSH) to oxidized (GSSG) glutathione. **J** WT and 4E KO MEF were grown as in (**C**), and total glutathione was measured. **K** WT and 4E KO MEF grown as in (**C**) were labelled with CM-DCFDA and analyzed by flow cytometry. **L** 4E KO MEF were grown in glucose starved medium (Glc strv) and treated with vehicle (V), N-acetyl cysteine (NAC) or Catalase (CAT) for 48 h. Cell death was measured by PI staining and flow cytometry. Data are shown as the mean ± SD. Statistics: unpaired one-sided Student's *t* test (**A**–**C**, **E**–**G**, **I**–**L**); n = 3 independent experiments for (**A**–**C**, **E**–**G**, **I**–**L**). Source data are provided as a Source Data file.

essential for cell survival upon glucose starvation[9], we examined the contribution of this metabolic pathway to 4EBP1/2 protective function under glucose starvation (Fig. 3B). Treatment of 4EBP1/2 deficient HEK293 cells and MEFs with TOFA, an inhibitor of ACC activity and fatty acid synthesis, was sufficient to reduce cell death under glucose starvation (Fig. 3C, D). In support of this, siRNA-mediated kd of fatty acid synthase (FASN), the NADPH consuming enzyme during fatty acid synthesis (Fig. 3B), rescued sh4EBP1/2 HEK293 or 4E KO MEFs from glucose starvation-induced cell death, confirming that the inhibition of fatty acid synthesis rescues 4EBP1/2 depleted cells from glucose starvation (Fig. 3E, F).

To further ascertain that 4EBP1/2 control fatty acid synthesis in response to glucose starvation, we measured the impact of 4EBP1/2 depletion on fatty acid synthesis activity using $^{14}$C acetate labeling to quantify $^{14}$C incorporation into the cellular lipid fraction (Fig. 3G). Indeed, glucose starvation led to reduced $^{14}$C-labelled lipids in control shScr HEK293 cells and WT MEFs, which was not observed in sh4EBP1/2 HEK293 cells and 4E KO MEFs (Fig. 3H, I). The 4EBP1/2 deficient cells displayed higher amounts of $^{14}$C-labelled lipids than corresponding control cells under glucose starvation, indicating that 4EBP1/2 are essential for cells to inhibit fatty acid synthesis activity in response to glucose starvation. We next verified that such regulation impinges on fatty acid oxidation, as would be expected since malonyl-CoA accumulation blocks fatty acid oxidation and that fatty acid synthesis and fatty acid oxidation are controlled by glucose availability. Using [$^3$H] palmitate labeling to measure rates of fatty acid oxidation, we observed that under glucose deprivation 4EBP1/2 loss restricted fatty acid oxidation compared to control cells (Fig. S4B–D), indicating that 4EBP1/2 promote fatty acid oxidation in response to glucose starvation likely as a consequence of blocking fatty acid synthesis.

We next assessed the importance of 4EBP1/2-mediated regulation of fatty acid synthesis in preserving NADPH levels and maintaining redox balance in response to glucose deprivation. Notably, inhibition of fatty acid synthesis with TOFA led to a significant elevation of the NADPH/NADP$^+$ ratio in 4EBP1/2 deficient cells upon glucose withdrawal (Fig. 3J, K), coupled with reduced endogenous H$_2$O$_2$ levels (Fig. 3L, M). Altogether, our findings demonstrate that during glucose starvation, 4EBP1/2 promote cell viability by inhibiting fatty acid synthesis to preserve NADPH levels and reduce ROS.

### 4EBP1/2 selectively regulate the translation of *ACACA* to preserve cell viability under glucose deprivation

We next sought to determine the mechanism by which 4EBP1/2 regulate fatty acid synthesis in response to glucose starvation. Since our proteomics analysis indicated that the expression of ACC1 is deregulated in 4EBP1/2 deficient HEK293 cells (Fig. 3A), and since 4EBP1/2 is known to regulate a subset of transcripts[27,28,33,34], we hypothesized that 4EBP1/2 selectively control the translation of *ACACA* (gene encoding ACC1) under glucose deprivation. We initially confirmed by immunoblotting that ACC1 levels were impacted upon 4EBP1/2 loss. Expression of ACC1 was consistently higher in 4EBP1/2 deficient MEFs and HEK293

cells compared to corresponding controls under glucose starved conditions (Fig. 4A, B). Moreover, we observed a more rapid decline in ACC1 expression upon glucose shortage in control cells compared to 4EBP1/2 deficient cells (Fig. 4A, B). The levels of phospho-ACC1 showed a similar trend than total ACC1 (Fig. 4A, B), and given our observation that AMPK is not more active in 4EBP1 deficient cells (Fig. S2G, H), we expect that this is concordant with total ACC1 expression rather than AMPK-mediated phosphorylation of ACC1. Additionally, 4EBP1$^{AA}$ overexpression led to a reduction of ACC1 levels in HeLa cells under glucose deprivation (Fig. S5A). None of the other enzymes of the fatty acid synthesis pathway (Fig. 3B), including ATP citrate lyase (ACLY), ACC2 and FASN, were consistently impacted upon 4EBP1/2 loss under glucose starvation (Fig. 4A, B).

Following the above, we next examined whether 4EBP1/2 control *ACACA* translation in response to glucose starvation. We firstly verified that transcriptional regulation is not involved, since *ACACA* mRNA levels were unchanged between control and 4EBP1/2 deficient HEK293 cells under glucose deprivation (Fig. 4C). In contrast, by quantifying levels of *ACACA* transcripts in polysomal and total mRNA, and by calculating translation efficiency as the ratio of polysomal to total mRNA levels[35], we found that translation efficiency of *ACACA* transcript was significantly higher in sh4EBP1/2 HEK293 cells compared to control cells under glucose deprivation (Fig. 4D). This supports that 4EBP1/2 may inhibit *ACACA* translation upon glucose withdrawal.

Given that 4EBP/eIF4E-mediated selective translational control is mediated through the 5′UTR of each target, we investigated the potential regulation of *ACACA* 5′UTR by 4EBP1/2 in response to glucose starvation. Since *ACACA* encodes several isoforms harboring different 5′UTR[36], we focused on the 5′UTR present in human *ACACA* transcript variant 3 and conserved in mice, as it is the most highly expressed isoform in HEK293 cells (data not shown). Notably, using a luciferase reporter, we observed that *ACACA* 5′UTR activity was significantly decreased upon glucose starvation in control MEFs and HEK293 cells (Fig. 4E, F). More importantly, *ACACA* 5′UTR activity was higher in 4E KO MEFs and sh4EBP1/2 HEK293 cells under glucose starvation compared to respective control cells (Fig. 4E, F).

To determine the contribution of the *ACACA* 5′UTR to the 4EBP1/2-mediated regulation of ACC1 expression, we ectopically expressed HA-tagged *ACACA*, with or without flanking by the 5′UTR (Fig. 4G), and monitored the impact of 4EBP1/2 on exogenous ACC1 protein levels during glucose starvation. While the expression of 5′UTR-absent ACC1 (ACC1-HA) did not differ between sh4EBP1/2 and control HEK293 cells under glucose starvation, the level of 5′UTR-containing ACC1 (UTR-ACC1-HA) was higher in sh4EBP1/2 cells as compared to controls during glucose starvation (Fig. 4G). Altogether, these data support that the *ACACA* 5′UTR plays a role in the 4EBP1/2-mediated inhibition of *ACACA* translation under glucose-deprived conditions.

To determine the contribution of increased ACC1 expression towards the sensitivity of 4EBP1/2 deficient MEFs and HEK293 cells to glucose starvation, we assessed the impact of ACC1 kd on cell viability during glucose starvation. We found that ACC1 kd rescued 4EBP1/2

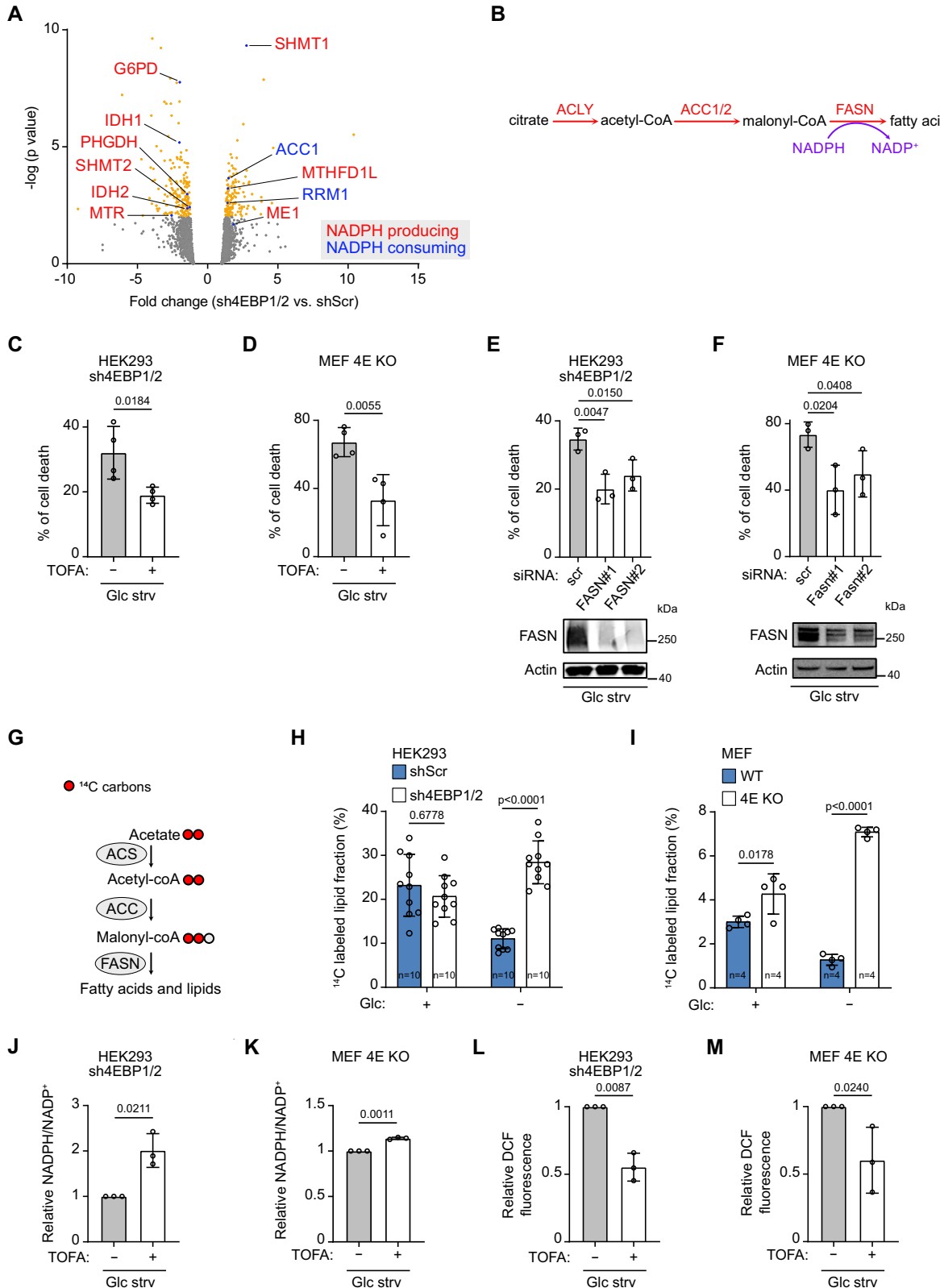

deficient cells from glucose starvation-induced cell death (Fig. 4H, I), in line with our findings using an ACC inhibitor (Fig. 3C, D), and was accompanied with increased NADPH/NADP⁺ ratio in 4EBP1/2 KO MEFs (Fig. S5B). Conversely, overexpression of ACC1 in HEK293 cells resulted in increased rates of cell death and reduced NADPH/NADP⁺ ratio during glucose starvation (Fig. S5C, D). These results highlight that 4EBP1/2 protect cells from glucose starvation by inhibiting the translation of *ACACA* in a 5'UTR dependent manner, thus reducing fatty acid synthesis, preserving NADPH, and limiting oxidative stress.

## 4EBP1/2 promote oncogenic transformation by controlling ACC1 level and mitigating oxidative stress

The cellular response to glucose starvation is closely linked to oncogenic transformation and tumorigenicity as matrix detachment, a

**Fig. 3 | 4EBP1/2 control fatty acid synthesis activity in response to glucose starvation to preserve redox balance and protect cells. A** Control (shScr) and sh4EBP1/2 HEK293 cells were grown in complete medium or glucose (Glc) starved for 30 h and proteomics was analyzed by MS. Differentially expressed proteins in shScr versus sh4EBP1/2 HEK293 cells under glucose starvation (*p*-value < 0.05) correspond to yellow and blue dots. Proteins involved in NADPH producing or NADPH consuming processes are highlighted. **B** Scheme of the fatty acid synthesis pathway highlighting the enzymatic steps and consumption of NAPDH. ACLY: ATP citrate lyase, ACC1/2: acetyl-CoA carboxylase 1/2, FASN: fatty acid synthase. **C**, **D** The indicated cell lines were grown in glucose starved medium (Glc strv) with or without TOFA for 48 h. Cell death was measured by PI staining and flow cytometry. **E**, **F** The indicated cell lines were transfected with control siRNA (scr) or siRNAs targeting *FASN* and glucose starved (Glc strv) for 48 h. Cell death was analyzed as in (**C**, **D**). FASN protein levels were analyzed by immunoblotting. **G** Scheme of the [$^{14}$C] acetate labeling assay to measure fatty acid synthesis activity. ACS: acetyl-CoA synthetase, ACC: acetyl-CoA carboxylase, FASN: fatty acid synthase. **H, I** The indicated cell lines grown in complete medium or glucose (Glc) starved for 24 h were labeled with [$^{14}$C] acetate in the last 18 h. [$^{14}$C] was measured in the lipid fraction and normalized to total protein levels. In (**H**), $n = 10$ biological replicates, and in (**I**), $n = 4$ biological replicates. (**J, K**) The indicated cell lines were glucose starved (Glc strv) for 24 h with or without TOFA, and NADP$^+$ and NADPH levels were measured. (**L, M**) The indicated cell lines were glucose starved (Glc strv) for 24 h with or without TOFA were labelled with CM-DCFDA and analyzed by flow cytometry. Data are shown as the mean ± SD. Statistics: unpaired one-sided Student's *t* test (**C**–**F**, **J**–**M**), two way ANOVA (**H, I**); $n = 3$ independent experiments for (**C**–**F**, **J**–**M**). Source data are provided as a Source Data file.

hallmark of transformation and tumorigenicity, triggers glucose starvation-like energetic stress, characterized by ATP depletion and elevated ROS[37]. Furthermore, mechanisms mediating the adaptation to glucose deprivation, such as through the mitigation of oxidative stress and control of fatty acid synthesis, are also required to support oncogenic transformation and tumorigenicity[9,37,38]. Following our findings that 4EBP1/2 exert pro-survival functions in response to glucose starvation, we sought to explore the contribution of 4EBP1/2 to oncogenic transformation.

While 4EBP1/2 have been reported to be essential for oncogenic RAS transformation of primary fibroblasts[39], it is not known whether 4EBP1/2 support transformation by other oncogenes or whether they contribute to the maintenance of the oncogene-transformed state, as is expected if 4EBP1/2 are indeed critical factors to promote survival during energetic stress[9]. Using soft agar colony formation assays, we uncovered that 4EBP1/2 is indeed necessary for HER2 transformation of mouse mammary epithelial cells (NT2197) in vitro (Fig. 5A). Similarly, 4EBP1/2 kd restricted the ability of KRAS$^{V12}$-transformed, immortalized NIH 3T3 fibroblasts to form colonies in soft agar (Fig. S6A). Conversely, overexpression of 4EBP1$^{AA}$ in HeLa cells led to a significant increase in colony formation in soft agar compared to control (EV) (Fig. 5B). Thus, these data demonstrate that the pro-tumorigenic functions of 4EBP1/2 are neither restricted to the *RAS* oncogene nor only to the initiation of cellular transformation.

To determine how 4EBP1/2 deficiency inhibits oncogenic transformation, we assessed the possible involvement of oxidative stress and uncontrolled fatty acid synthesis. Notably, treatment of 4E KO NT2197 cells with antioxidants—CAT, NAC or TROLOX—or with the ACC inhibitor TOFA rescued colony formation in soft agar (Fig. 5C), while these had no observable effects on 4EBP1/2 WT NT2197 cells (Fig. S6B). Similarly, antioxidant treatment restored colony formation of NIH 3T3 KRAS$^{V12}$ 4EBP1/2 kd cells in soft agar (Fig. S6C). Importantly, we found that clustered regularly interspaced short palindromic repeats interference (CRISPRi)-mediated kd of Acc1 expression was sufficient to restore colony formation in 4EBP1/2 deficient NT2197 cells (Fig. 5D). Thus, we conclude that 4EBP1/2 support oncogenic transformation at least in part by negatively regulating ACC1 and oxidative stress.

We then sought to recapitulate these findings in vivo and found that 4E KO NT2197 cells were unable to form any palpable tumors when injected in the fat pad of immunocompromised mice, in sharp contrast to WT NT2197 cells, which formed sizeable tumors in all mice (12/12) (Fig. 5E, F). We verified that 4EBP1 reexpression in 4E KO NT2197 cells restored tumor growth (Fig. S6D, E). In line with this, over-expression of 4EBP1$^{AA}$ in HeLa cells promoted tumor growth relative to control (EV) HeLa cells when injected into the flanks of immunocompromised mice, which was not the case with overexpression of the eIF4E-non-binding mutant 4EBP1$^{AA, YL}$ (Fig. 5G, H). To ascertain the contribution of ACC1 to the observed phenotype of NT2197 tumors in vivo, we assessed the impact of targeting ACC1 expression on the growth of 4E KO NT2197 tumors. Acc1 kd (shAcaca) in 4E KO NT2197

cells led to a major increase of tumor mass as compared to control (shGFP 4E KO NT2197) tumors (Fig. 5I, J). Interestingly, analysis of protein oxidation in NT2197 tumors, by dityrosine immunoblot, indicated increased levels of oxidative stress upon 4E KO, when compared to WT tumors, which was prevented with Acc1 kd (Fig. S6F, G). Collectively, these data support a model wherein 4EBP1/2 promote oncogenic transformation, tumorigenicity and survival during glucose starvation through a common mechanism that entails reduced ACC1 expression to restrain fatty acid synthesis and, consequently, oxidative stress.

## 4EBP1 is clinically relevant and functional in brain tumors

Having found that 4EBP1/2 promote survival upon glucose starvation, a condition commonly encountered in solid tumors, as well as oncogenic transformation, we further investigated the clinical relevance of 4EBP1/2 in cancer. It has been reported that *EIF4EBP1* is overexpressed in numerous tumor entities of TCGA and in a pan-cancer analysis of TCGA data, high *EIF4EBP1* expression is associated with poor patient outcome[40]. Furthermore, analysis of TCGA and GTEx datasets indicates that, together with our previous report[41], *EIF4EBP1* is overexpressed in 17 different tumor types compared to corresponding normal tissues (Fig. S7A). In contrast, *EIF4EBP2* is only overexpressed in 3 out of these 17 tumor types in TCGA datasets (Fig. S7B), leading us to focus our analyses on *EIF4EBP1*. Indeed, we uncovered that high *EIF4EBP1* expression correlated with significantly decreased overall survival in three different tumor types (Fig. 6A, Fig. S7C, D), including glioblastoma, highlighting *EIF4EBP1* expression as a potential prognostic biomarker in these tumor entities. Within this context, it is worth mentioning that glucose levels are low in the interstitial compartment of the brain as compared with blood[42,43].

To survive the pre-existing low glucose microenvironment of the brain, glioma tumor cells or cancer cells that metastasize to the brain are forced to acquire resistance to glucose starvation and/or use alternative energy sources[44,45]. Thus, we turned our attention toward malignant gliomas, the most common form of brain tumor and typically characterized by glucose deprivation[46]. Within different glioma types, *EIF4EBP1* expression is found to be higher in the most aggressive CNS World Health Organisation (WHO) grade 4 glioblastoma, followed by grade 3 and grade 2 gliomas (Fig. 6B). Additionally, we analyzed proteomic data obtained from glioblastoma patient samples and observed that 4EBP1 protein is overexpressed in glioblastoma tissues compared to non-tumorigenic brain tissues (NTBT) (Fig. S7E). Importantly, 4EBP1 protein levels were found to be negatively correlated with ACC1 protein expression in glioblastoma (Fig. 6C), supporting our model in which 4EBP1 represses ACC1 synthesis.

To functionally dissect the role of 4EBP1 in glioma, we analyzed the impact of 4EBP1 kd on the tumorigenic potential of human and mouse glioma cells, U-87 MG and GL-261, respectively. We first confirmed that 4EBP1 kd sensitizes such glioma cells to glucose starvation-induced cell death (Fig. S7F, G), and that inhibiting protein synthesis or ROS rescued 4EBP1 kd glioma cells from glucose starvation-induced

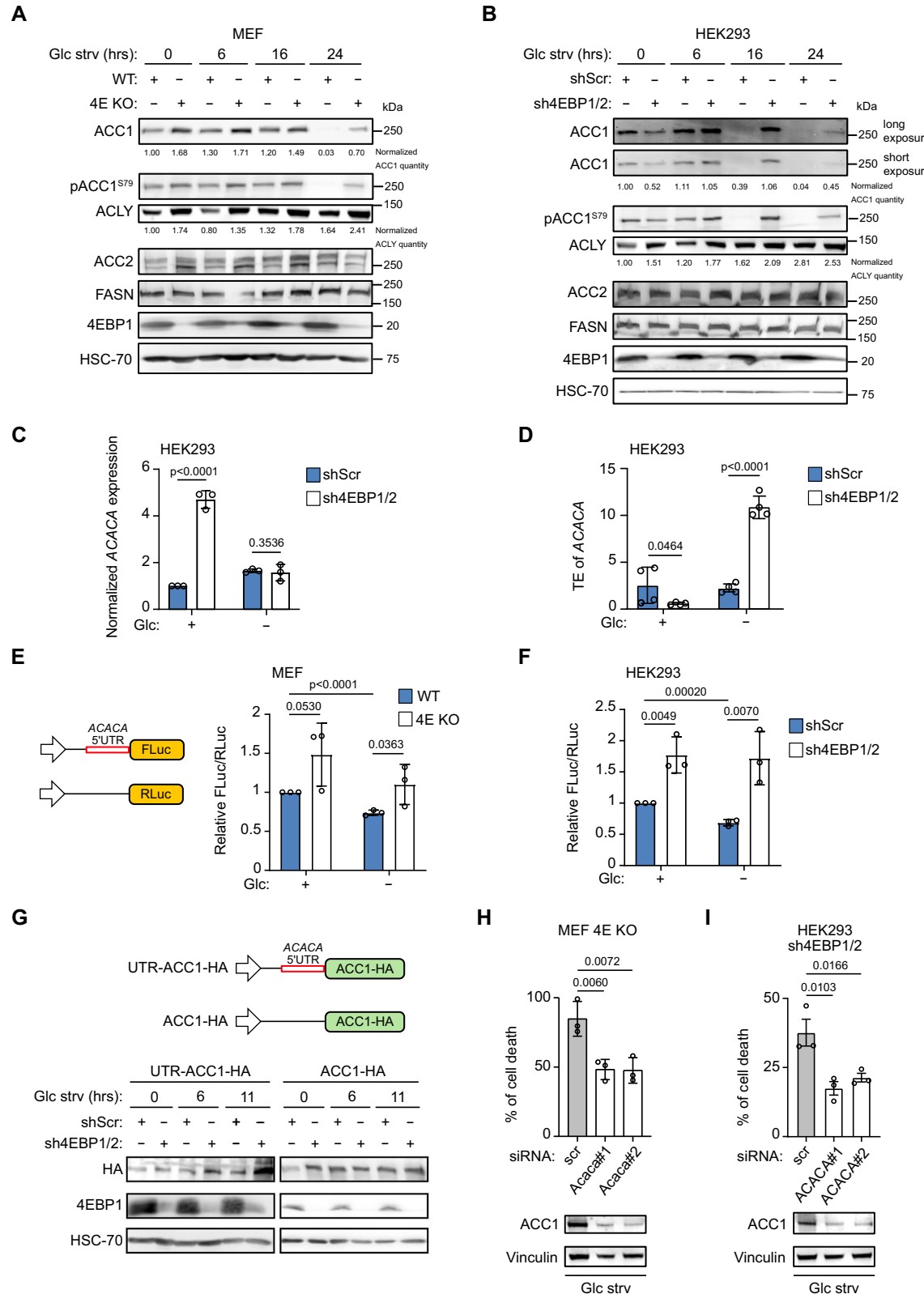

cell death (Fig. S7F). Moreover, 4EBP1 kd severely restricted the ability of glioma cells to form colonies in soft agar (Fig. 6D, Fig. S7H), as previously demonstrated with other cell lines. Importantly, inhibiting ACC with TOFA or supplementing cells with antioxidants rescued colony formation of 4EBP1 deficient glioma cells (Fig. 6E, Fig. S7I–K), which was not observed with corresponding 4EBP1 proficient cells (Fig. S8A, B). These data further corroborate that in glioma cells, 4EBP1

promotes tumorigenicity in vitro by means of controlling redox balance and fatty acid synthesis.

We next evaluated pro-tumorigenic functions of 4EBP1 in glioma cells in vivo by first injecting control (shScr) and 4EBP1 kd U-87 MG cells into the flanks of NOD *SCID* mice and observed that 4EBP1 depleted cells formed significantly smaller tumors as compared to controls (Fig. 6F). Of note, 4EBP1 kd U-87 MG tumors displayed

**Fig. 4 | 4EBP1/2 repress ACACA translation under glucose starvation. A, B** WT and 4E KO MEF (**A**), or shScr and sh4EBP1/2 HEK293 cells (**B**) were grown in complete medium or glucose starved (Glc strv) for the indicated times, and analyzed by immunoblotting using antibodies against the indicated proteins. Representative results of two independent experiments are shown. **C** ShScr and sh4EBP1/2 HEK293 cells were grown in complete medium or glucose (Glc) starved for 16 h, and *ACACA* mRNA expression was analyzed by qRT-PCR. **D** ShScr and sh4EBP1/2 HEK293 cells were grown in complete medium or glucose (Glc) starved for 6 h, and translation efficiency (TE) of *ACACA* mRNA was calculated by measuring the levels of polysomal and total *ACACA* mRNA by qRT-PCR. *n* = 4 independent experiments. **E, F** WT and 4E KO MEF (**E**), or shScr and sh4EBP1/2 HEK293 cells (**F**) were transfected with an *ACACA* 5′UTR-containing Firefly Luciferase construct and a control *Renilla*

Luciferase vector. Cells were grown in complete medium or glucose (Glc) starved for 6 h, and luminescence was measured. **G** HEK293 cells were transfected with an HA-tagged ACC1 expressing vector containing or not the *ACACA* 5′UTR. Cells were grown in complete medium or glucose starved (Glc strv) for the indicated times, and analyzed by immunoblotting using antibodies against the indicated proteins. Representative results of three independent experiments are shown. **H, I** 4E KO MEF (**H**) or sh4EBP1/2 HEK293 cells (**I**) were transfected with control siRNA (scr) or siRNAs targeting *ACACA* and grown in glucose starved medium (Glc strv) for 48 h. Cell death was measured by PI staining and flow cytometry. ACC1 protein levels were analyzed by immunoblotting. Data are shown as the mean ± SD. Statistics: unpaired one-sided Student's *t* test (**C–F, H, I**); *n* = 3 independent experiments for (**C, E, F, H, I**). Source data are provided as a Source Data file.

increased 8-Oxo-2′-deoxyguanosine (8-oxo-dG) staining as compared to control tumors (Fig. S8C, D), supporting the hypothesis that 4EBP1 curbs oxidative stress in tumors. To determine whether 4EBP1 is also important for tumor maintenance, we used a doxycycline-inducible shRNA system to target 4EBP1 expression in established tumors. Namely, engineered inducible 4EBP1 kd U-87 MG cells were injected into the flanks of NOD *SCID* mice, and once tumor size reached 100 mm³, 4EBP1 kd was induced by adding doxycycline to drinking water. Consequently, we observed an inhibition of tumor growth in doxycycline treated mice harboring sh4EBP1 U-87 MG tumors but not in shScr U-87 MG tumors or tumors in mice unexposed to doxycycline (Fig. 6G, Fig. S8E, F). These data suggest that 4EBP1 maintains the growth of established glioma tumors in vivo. To further corroborate our findings that 4EBP1 supports glioma tumor growth, specifically when localized in the brain, we performed orthotopic injection of control and sh4EBP1 U-87 MG cells. While both control and 4EBP1 deficient cells generated tumors, mice bearing sh4EBP1 U-87 MG tumors survived longer as compared with controls (Fig. 6H).

Finally, to assess 4EBP1 function in an immunocompetent mouse model, we injected control and sh4EBP1 GL-261 cells into the brains of C57WT mice. Mice carrying sh4EBP1 GL-261 tumors showed a significant extension of survival compared to mice with control (shScr) tumors (Fig. 6I), suggesting that 4EBP1 promotes glioma aggressiveness even in the presence of a functional immune system. Importantly, Acc1 kd enhanced tumor aggressiveness of orthotopically injected sh4EBP1 GL-261 cells, as evidenced by reduced mice survival (Fig. 6J). This supports a model whereby 4EBP1 function is exploited by glioma cells to reduce Acc1 expression to promote tumor aggressiveness. Collectively, our data highlight that *EIF4EBP1* is linked to malignancy and poor outcome in various human tumor types, including glioma, and that 4EBP1 exerts a pro-tumorigenic function in glioblastomas by reducing ACC1 expression.

## Discussion

Glucose starvation represents a physiological stress that necessitates coordinated cellular responses to prevent cell death[4]. The activity of mTORC1 is critical for the response to glucose depletion, such that mTORC1 inhibition is required for cell survival under this condition[12,15]. Our data illustrate for the first time that downstream of mTORC1, 4EBP1/2 are responsible for promoting cell survival during glucose starvation. It was previously reported that the protective effect exerted by mTORC1 inhibition does not entail oxidative phosphorylation nor autophagy[15], but is thought to occur due to mTORC1 regulation of p53 translation[47], thus preventing apoptosis when mTORC1 is shut down. Instead, we observed that 4EBP1/2 protect even p53 deficient cells, including 4EBP1/2 WT MEFs −which are KO for p53− and Kelly neuroblastoma cells that carry an inactivating *TP53* mutation, suggesting rather that 4EBP1/2 protect cells during glucose starvation independently of p53.

Additionally, it was proposed that mTORC1 inhibition supports cell survival under glucose starvation by reducing ATP consumption[13,15]. Given the fact that pharmacological inhibition of total protein synthesis rescued glucose-starved cells that harbor

constitutively active mTORC1[15], and given that protein synthesis is the most ATP-consuming cellular process[48], it was assumed that mTORC1 inhibition protects cells from glucose starvation by restricting overall protein synthesis[15]. Interestingly, our data do not lend support to such a model, as we report that 4EBP1/2 protective function does not rely on the regulation of overall protein synthesis or of ATP levels. Furthermore, while other reported functions of 4EBP1/2 include restricting cell proliferation and mitochondrial activity[27,28,49], we found that this was not the case under glucose starvation. On the contrary, our data support that 4EBP1/2 act as pro-survival factors under glucose deprivation, which is not the case under serum starvation or pharmacological inhibition of mTORC1, as previously reported[27]. This highlights that the type of stress encountered by cells may dictate 4EBP1/2 cellular functions.

The cellular response to glucose starvation encompasses profound metabolic reprogramming, during which anabolic processes are blocked and catabolic processes are activated[4]. MTORC1 represents a major regulator of such a metabolic switch in response to glucose availability, as it controls protein, lipid and nucleotides synthesis, as well as autophagy[7,8].

Our findings uncovered that downstream of mTORC1, 4EBP1/2 are key mediators of the metabolic switch induced by glucose withdrawal by reprogramming lipid metabolism and especially restricting fatty acid synthesis activity. This allows cells to preserve NADPH levels and maintain cellular redox balance, similar to the reported function of the energy sensor AMPK[9]. However, unlike AMPK, which regulates fatty acid synthesis at the posttranslational level, i.e. by phosphorylating and inhibiting the fatty acid synthesis rate limiting enzyme ACC1, we found that 4EBP1/2 control this process at the translational level, highlighting 4EBP1/2 as previously unrecognized translational regulators that fine tune redox balance according to the intracellular energy state. This reveals an unexpected, non-transcriptional mechanism of mTORC1 in controlling of fatty acid synthesis[50].

The ability of 4EBP1/2 to modulate metabolic processes, such as proliferation and mitochondrial activity, relies on their ability to selectively restrict the translation of hundreds of specific transcripts[27,28,33,34]. In this context, we uncovered that in response to glucose starvation, 4EBP1/2 restrain fatty acid synthesis by selectively inhibiting the synthesis of ACC1. It was previously reported that eIF4E selectively promotes *ACACA* translation in T cells[51] and in liver tissue of mice fed with a high-fat diet[52]. In particular, the transition of CD4 + T cells from quiescence to activation, which metabolically mirror changes in glucose availability, is driven by eIF4E-promoted *ACACA* translation, dependent upon a 5′UTR of *ACACA*[51]. Similarly, we report that a second 5′UTR of *ACACA* supports 4EBP1/2-mediated control of *ACACA* translation, highlighting that this *ACACA* 5′UTR represents a genetic element linking fatty acid synthesis activity to the energetic state of the cell. Taken together, the regulation of cell metabolism by 4EBP1/2, through inhibition of fatty acid synthesis, proliferation[27] and mitochondrial activity[28], is compatible with 4EBP1/2 acting as stress-responsive metabolic switches, following mTORC1 inhibition, to steer cells towards a more quiescent or low energy state. Since our

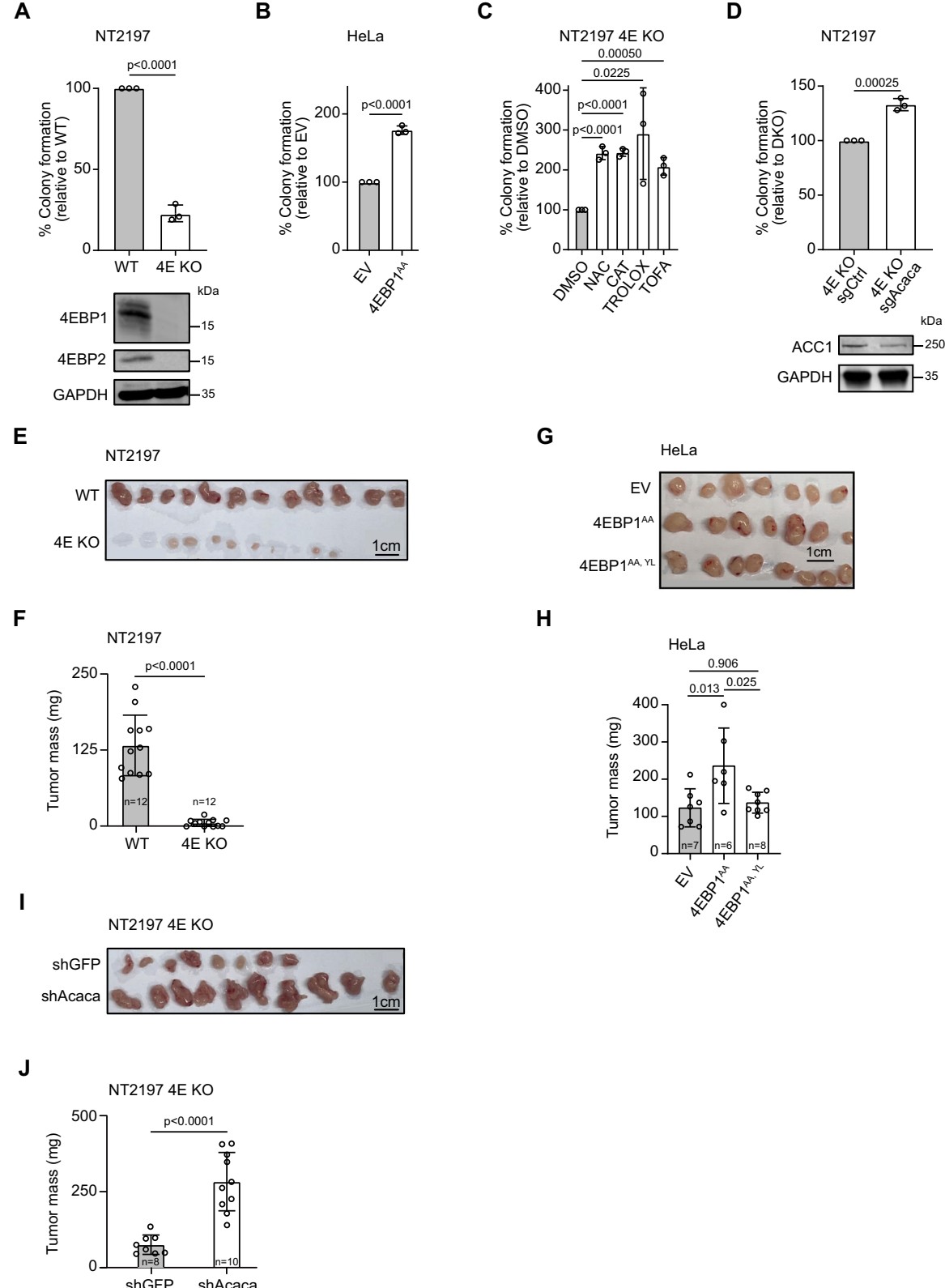

proteomic data showed dysregulated expression of other enzymes of NADPH metabolic pathways, and in particular G6PD, we can not exclude that regulation of pentose phosphate pathway activity contributes to the pro-survival functions of 4EBP1/2 during glucose starvation.

Altogether, our data support a model whereby in addition to AMPK-mediated phosphorylation of ACC1, cells evolved a parallel

mechanism to inhibit ACC1 in response to energetic stress through mTORC1-4EBP1/2-mediated translational repression of *ACACA* mRNA. We speculate that these represent two distinct but complementary mechanisms, given that phosphorylation occurs rapidly and is easily reversible by the action of phosphatases, while repression of mRNA translation occurs more slowly but may serve as a more durable and efficient means of blocking ACC1. Notably, in bacteria, ACC is also

**Fig. 5 | 4EBP1 supports oncogenic transformation in vitro and in vivo. A, B** WT and 4EBP1/4EBP2 DKO (4E KO) NT2197 (**A**), or empty vector (EV) and 4EBP1$^{AA}$ expressing HeLa cells (**B**) were grown in soft agar for 21 days. Colonies and single cells were counted, and colony formation efficiency was calculated and normalized to respective control. Protein expression of 4EBP1 and 4EBP2 was analyzed by immunoblotting. **C** 4E KO NT2197 cells were grown in soft agar for 21 days and treated with DMSO, NAC, CAT, TROLOX or TOFA. Colonies and single cells were counted, and colony formation efficiency was calculated and normalized to DMSO. **D** Control (sgCtrl) and ACC1 targeting CRISPRi (sgAcaca) 4E KO NT2197 cells were grown in soft agar for 21 days. Colony formation efficiency and proteins level were analyzed as in (**A, B**). $n = 3$ independent experiments for (**A–D**). **E, F** WT or 4E KO NT2197 cells were injected in the mammary fat pad of NOD *SCID* gamma mice. Tumors were harvested, photographed (**E**) and weighed (**F**). $n = 12$ mice per cell line. **G, H** EV, 4EBP1$^{AA}$ or 4EBP1$^{AA, YL}$ expressing HeLa cells were injected in the flank of NOD *SCID* gamma mice. Tumors were harvested, photographed (**G**) and weighed (**H**). $n = 6–8$ mice per cell line. **I, J** Control (shGFP) or stable ACC1 knock down (shAcaca) 4E KO NT2197 cells were injected in the flank of NOD *SCID* gamma mice. Tumors were harvested, photographed (**I**) and weighed (**J**). $n = 8–10$ mice per cell line. Data are shown as the mean ± SD. Statistics: unpaired one-sided Student's $t$ test (**A–D**), two way ANOVA (**F, H, J**). Source data are provided as a Source Data file.

regulated at both the translational and post-translational levels in response to glucose availability[53], in agreement with the possibility that both modes of ACC1 regulation are evolutionarily conserved.

While mTORC1 represents a promising therapeutic target in cancer, clinical trials assessing mTOR inhibitors as monotherapy have not been so promising[54]. This may be due to the dual role of mTORC1 in cancer, one as an anabolic driver when active and another as a metabolic break when inhibited. This was well illustrated by the effects of rapamycin on pancreatic cancer growth, as rapamycin treatment led to inhibition of proliferation in well perfused regions of the tumor, but also promoted cell survival in poorly vascularized regions of the same tumor[55]. In keeping, the role of 4EBP1 in cancer remains similarly unclear[20]. While 4EBP1 exhibits a tumor suppressive function in mouse models of lymphoma, head and neck squamous cell carcinoma and prostate cancer[56,57], 4EBP1 KO mice do not develop tumors per se, thus excluding 4EBP1 as a bona fide tumor suppressor[58]. Furthermore, 4EBP1 has been shown to exert pro-tumorigenic functions, as it is required for oncogenic RAS transformation[39], promotes breast cancer development in vivo[59], and correlates with poor patient outcome in several tumor types at the transcriptional level[40]. Here, our data further support a pro-tumorigenic function of 4EBP1 in vitro and in vivo, as we demonstrate that 4EBP1 mediates HER2 transformation of mouse mammary epithelial cells and tumorigenicity of glioma cells.

The role of 4EBP1 in cancer is likely determined by the levels of metabolic stress present in tumors, such that 4EBP1 acts as a pro-tumorigenic factor within metabolically challenged tumor environments, as was previously proposed for AMPK[60–62]. In particular, glucose concentrations in the interstitial space of brain are low compared to plasma[42,43], and glioblastoma are characterized by further reduction of glucose levels in the central region of the tumor[46]. Indeed, acquiring resistance to glucose starvation is essential for glioma cells and for breast tumor cells metastasizing to the brain[44,45]. With this perspective, and based on our data, we propose that 4EBP1 confers glioma cells the ability to adapt to such metabolic stress by preserving cellular redox balance and restricting ACC1 expression, by co-opting the mechanisms of 4EBP1 function in response to glucose deprivation. This is in line with the proposed function of AMPK in mediating cell survival under glucose starvation and tumorigenesis through inhibition of ACC1 and prevention of oxidative stress[9]. Therefore, 4EBP1 represents a metabolic regulator exploited by cancer cells to adapt to the adverse conditions of the tumor microenvironment.

It is worth noting that since 4EBP1 is post-translationally inhibited by mTORC1, which is overactive in numerous cancers, and as evidenced by increased levels of phosphorylated 4EBP1 reported in various tumor tissues, it is assumed that 4EBP1 is inactive in tumors[20]. However, the amount of total 4EBP1 protein, which is also a contributing factor, is rarely monitored. Interestingly, we observed that *EIF4EBP1* mRNA expression is tightly correlated with 4EBP1 phosphorylation level in glioblastoma (Fig. S8G–I), suggesting that in glioblastoma higher phosphorylated 4EBP1 results from *EIF4EBP1* overexpression and therefore does not indicate increases of inactive 4EBP1. Furthermore, the activity of 4EBP1, which reflects mTORC1 activity, was shown to be directly dependent on the proximity to blood vessels in glioblastoma, such that the highest 4EBP1 activity was detected in areas furthest from blood vessels, corresponding to oxygen and glucose deprived areas[63]. This raises the possibility that upregulation of *EIF4EBP1*, as observed in numerous cancer types[40], leads to increased 4EBP1 activity in metabolically challenged tumor areas. It is also worth noting that oncogenic transcription factors, such as ETS1, MYBL2, MYC and MYCN[41,64,65] promote *EIF4EBP1* overexpression, further supporting the clinical relevance of *EIF4EBP1* as a pro-tumorigenic gene. In synopsis, our findings suggest that 4EBP1 may represent a therapeutic target in metabolically challenged tumor types, while warranting caution on the use of mTOR inhibitors in these cancers. In addition, our findings reveal that the mTORC1-4EBP1/2 axis inhibits fatty acid synthesis during glucose starvation and that this particular function is exploited by tumor cells for their own selective advantage.

## Methods

### Reagents and antibodies

Reagents used for this study were as follows: cycloheximide (CHX), N-acetylcysteine (NAC), Catalase (CAT), 5-(Tetradecyloxy)−2-furoic acid (TOFA), 6-hydroxy-2,5,7,8-tetramethylchroman-2-carboxylic acid (TROLOX) and gemcitabine hydrochloride were from Sigma-Aldrich. Doxycycline hydrochloride (DOX) was from Santa Cruz. Rapamycin, KU-0063794 and 4EGI-1 were from Selleckchem.

Antibodies used in this study were as follows: anti-4E-BP1 (Cell Signaling Technology, 9644, 1:1000), anti-4E-BP2 (Cell Signaling Technology, 2845, 1:500), anti-ACC1 (Cell Signaling Technology, 4190, 1:500), anti-ACC2 (Cell Signaling Technology, #8578, 1:1000), anti-β-Actin (Sigma Aldrich, A2228, 1:1000), anti-GAPDH (Cell Signaling Technology, 2118, 1:2000), anti-LC3B (Cell Signaling Technology, 2775, 1:1000), anti-phospho-Acetyl-CoA Carboxylase (S79) (Cell Signaling Technology, 3661, 1:500), anti-phospho-AMPKalpha (T172) (Cell Signaling Technology, 2535, 1:500), anti-phospho-S6 Ribosomal Protein (S240/244) (Cell Signaling Technology, 2215, 1:1000), anti-Vinculin (Cell Signaling Technology, 4650, 1:1000), anti-phospho-ULK1 (S555) (Cell Signaling Technology, 5869, 1:1000), anti-eIF4E (Cell Signaling Technology, 9742, 1:1000), anti-FASN (Cell Signaling Technology, 3180, 1:1000), anti-ACLY (Cell Signaling Technology, 13390, 1:1000), anti-AMPKalpha (Cell Signaling Technology, 2532, 1:1000), anti-ULK1 (Cell Signaling Technology, 8054; 1:1000), anti-HSC-70 (Santa Cruz, sc-7298, 1:5000), mouse anti-HA-tag (Santa Cruz, sc-7392, 1:1000), anti-dityrosine (AdipoGen, JAI-MDT-020P, 1:1000), and anti-8-hydroxy-2'-deoxyguanosine (R&D Systems, 4354-MC-050, 1:250 for IHC). Anti-mouse IgG, HRP-linked (Cell Signaling Technology, 7076, 1:10000), anti-rabbit IgG, HRP-linked (Cell Signaling Technology, 7074, 1:10000), biotinylated goat anti-mouse IgG(H + L) (Abcam, ab64255, 1:200 for IHC), IRDye® 800CW goat anti-mouse IgG secondary antibody (LI-COR Bioscience, 925-32210, 1:10000), and IRDye® 800CW goat anti-rabbit IgG secondary antibody (LI-COR Bioscience, 925-32211, 1:10000).

### Cell culture

Cells were maintained using standard tissue culture procedures in a humidified incubator at 37 °C with 5% CO₂ and atmospheric oxygen. Stable HEK293 (human, female) control (shScr) and knock down for 4EBP1/4EBP2 (sh4EBP1/2) cell lines, WT (p53$^{-/-}$) and 4EBP1/4EBP2

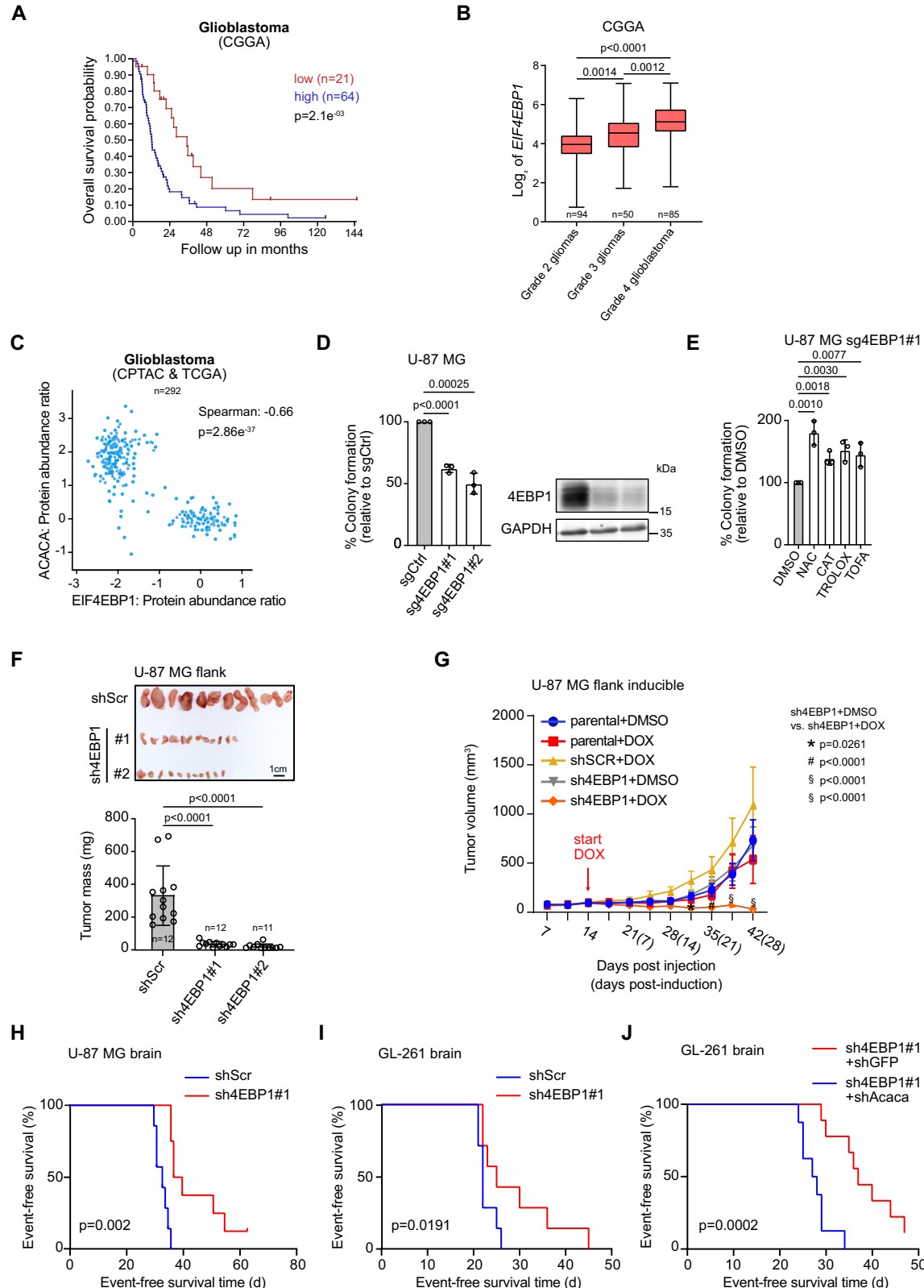

double knockout (DKO) (p53$^{-/-}$) MEFs (mouse, sex unspecified) were kind gifts from Prof. Nahum Sonenberg (McGill University, Canada). AMPKα+/+ and −/− MEFs (mouse, sex unspecified) were kindly provided by Dr. Keith Laderoute (SRI Biosciences, USA). NMuMG-NT2197 (mouse, female) (NT2197) control and 4EBP1/4EBP2 DKO cell lines were kind gifts from Dr. Ivan Topisirovic (McGill University, Canada). GL-261 glioma cell line (mouse, sex unspecified) was a kind gift from

Prof. Reuven Stein (Tel Aviv University, Israel). NIH 3T3 cells (mouse, male) stably expressing K-Ras$^{V12}$ have been previously described in ref. 21. Wild type HEK293, HEK293-T (human, female), HeLa (human, female), U-87 MG (human, male), MCF7 (human, female) cell lines were originally obtained from American Type Culture Collections (ATCC), and iPSC (human, female) were obtained from Takara Bio. Kelly (human, female) and IMR-32 (human, male) cells lines were generously

**Fig. 6 | 4EBP1 has clinical relevance in glioma and promotes glioma tumor-igenesis. A** Kaplan-Meier survival estimates of glioblastoma patients stratified by their *EIF4EBP1* mRNA levels (cut off first quartile) in the CGGA cohort. *p*-values were calculated using a log rank test. **B** Expression levels of *EIF4EBP1* per glioma grade in the CGGA cohort. Data were shown as boxplots with medians, interquartile ranges and lower/upper whiskers in. *p*-values were calculated using an unpaired and two-tailed parametric *t* test. **C** 4EBP1 protein levels plotted against the expression levels of ACC1 protein using CPTAC and TCGA GBM proteomic data. Co-expression level was quantified by calculating the Spearman's correlation coefficient. **D, E** The indicated cell lines were grown in soft agar for 21 days and were treated (**E**) or not (**D**) with the indicated compounds. *n* = 3 independent experiments. 4EBP1 protein levels were analyzed by immunoblotting (**D**). *p*-values were calculated using an unpaired and one-tailed Student's *t* test. **F** The indicated cell lines were injected in the flank of NOD *SCID* gamma mice. Tumors were harvested, photographed and weighed. *n* = 11–12 mice per cell line. *p*-values were calculated using two way ANOVA. **G** The indicated cell lines were injected in the flank of NOD *SCID* gamma mice. When tumors reached 100 mm³, mice were given doxycycline (DOX) or vehicle. Tumor volumes were measured at the indicated times. *n* = 12 mice per cell line. *p*-values were calculated using two way ANOVA. **H** ShScr (*n* = 7 mice) or sh4EBP1#1 (*n* = 8 mice) U-87 MG cells were injected intracranially in NOD *SCID* gamma mice. Survival of mice was monitored post injection. **I** ShScr (*n* = 10 mice) or sh4EBP1#1 (*n* = 10 mice) GL-261 cells were injected intracranially in C57WT mice. Survival of mice was monitored post injection. **J** sh4EBP1#1 containing shGFP (*n* = 9 mice) or shAcaca (*n* = 8 mice) GL-261 cells were injected intracranially in C57WT mice. Survival of mice was monitored post injection. p value was calculated using a log rank test for (**H–J**). Data are shown as the mean ± SD. Source data are provided as a Source Data file.

---

donated by Prof. Alexander Schramm (University Hospital Essen). Med8a (human, male) was a kind gift from Prof. Pablo Landgraf (University Hospital Cologne, Cologne), and HD-MB03 (human, male) cell line was generously donated by Prof. Till Milde (DKFZ, Heidelberg). NT2197 were cultured in Dulbecco's modified Eagle medium (DMEM) supplemented with 10% fetal bovine serum (FBS), 1% penicillin/streptomycin (pen/strep), 10 µg/ml insulin, and 20 mM HEPES, pH 7.5. NIH 3T3 K-Ras$^{V12}$ were cultured in DMEM supplemented with 10% bovine calf serum. HD-MB03, Kelly and IMR-32 cell lines were cultured in Roswell Park Memorial Institute (RPMI) medium supplemented with 10% FBS and 1% pen/strep. iPSC line was cultured in Biolaminin 521 LN (Biolamina AB)-coated plates containing mTeSR1 medium (Stem Cell Technologies). All other cell lines were maintained in DMEM supplemented with 10% FBS, 1% pen/strep. All cell lines were routinely confirmed to be mycoplasma-free using Venor®GeM Classic kit (Minerva Biolabs, Berlin, Germany). All human cell lines were authenticated by STR-profiling (Genomics and Transcriptomics Laboratory, Heinrich-Heine University, Germany).

### Yeast culture
Yeast strains (all isogenic to BY4742) were grown in complex medium containing 1% (w/v) yeast extract and 2% (w/v) peptone without (YP) or with (YPD) 2% glucose. To pour solid agar plates, 2% agar was added to medium. For dot spot assays, yeast strains were grown to an OD600 of approximately 1, washed and diluted in a series of fivefold dilutions before eventually being stamped on the corresponding agar plate before incubation at 30 °C or 37 °C for 3–5 days. For incubation in liquid complete YPD or glucose-free YP medium, suspensions were adjusted to an OD600 of 0.1 prior to incubation at 200 rpm at 30 °C. The OD600 was measured throughout the experiment with a spectrophotometer. For survival analysis, BY4742 control or *eap1Δ* yeast strains were incubated in liquid YP medium at an OD600 of 0.1 for 2 weeks at 30 °C shaking at 300 rpm prior to streaking serial dilutions onto complete YPD agar plates.

### Animal models
All mouse work was performed in accordance with the institutional animal care use committee and relevant guidelines at the Ben-Gurion University, with protocols 34-06-2016, 35-06-2016 and 59-08-2019E. C57BL/6 J (C57WT) and NOD *SCID* gamma *Prkdc$^{scid}$* mice were used. Both male and female mice from 5 to 8 weeks of age were used for all experiments in this study (see details in the table below). In a specific experiment all mice were from the same sex and same age. All mice were housed under specific-pathogen-free (SPF) condition at the Ben-Gurion University facility, kept at 20 °C, 50% humidity in a 14 h light/10 h dark cycle.

### Xenograft tumor models
For sub-cutaneous injection, cancer cells ($5 \times 10^6$–$1 \times 10^7$) were injected into the flank or mammary fat pad of mice. Tumors size was monitored using calipers. When tumors reached the maximum of allowed size, mice were sacrificed, tumors were excised and weighed. Each tumor was cut in half and either fixed in formaldehyde 4% or snap-frozen in liquid nitrogen. When an inducible system were used, mice received 10 mg/kg/day, such that 0.05 mg/mL of doxycycline was added to the drinking water twice a week.

For orthotropic/intracranial injection, cancer cells were engrafted into the mouse brain using a stereotactic device. At the end of the experiment, mice were sacrificed and their brains were excised.

The maximum tumor burden permitted according to BGU animal ethics committee is 1500 mm³. We did not exceed this limit in the current study.

### Glucose starvation of cell culture
Glucose starvation was performed with subconfluent cultures (~50% confluency). Full medium was replaced with DMEM or RPMI containing no glucose and no sodium pyruvate supplemented with 10% dialyzed FBS and 1 mM glucose. When indicated, cells were treated with either Rapamycin (100 nM), KU-0063794 (1 µM), 4EGI-1 (10 µM), CHX (2 µg/ml), NAC (3 mM), Catalase (400 U/ml), TOFA (5 µM) or gemcitabine (1 µM) at the time of medium replacement.

### Vectors for genetically manipulating cell lines
**shRNA expression plasmids.** To generate the shRNA expression vectors that were not commercially available, complementary oligonucleotides corresponding to shRNAs targeting mouse *Acaca* were custom cloned (Genewiz) into AgeI and EcoRI restriction sites of the pLKO.1-neo (gift from Sheila Stewart - Addgene plasmid #13425 [http://n2t.net/addgene:13425]; RRID:Addgene_13425). Complementary oligonucleotides corresponding to shRNAs targeting human *EIF4EBP1* were custom cloned (Genewiz) into AgeI and EcoRI restriction sites of the Tet-pLKO-puro (gift from Dmitri Wiederschain - Addgene plasmid #21915 [http://n2t.net/addgene:21915]; RRID:Addgene_21915) vectors. Cloned shRNA sequences can be found in supplementary table 4. PLKO.1-puro scramble shRNA was a gift from David Sabatini (Addgene plasmid# 1864 RRID:Addgene_1864) and pLKO.1-neo shGFP was a gift from Kevin Janes (Addgene plasmid# 72571; RRID:Addgene_72571). All other pLKO.1 lentiviral shRNA vectors were pLKO.1-puro based and were retrieved from the arrayed Mission TRC genome-wide shRNA collections purchased from Sigma-Aldrich Corporation.

**CRISPRi/Cas9 plasmids.** To construct CRISPRi/Cas9 targeting vectors, a non-targeting control single guide RNA (sgRNA), or sgRNAs targeting human *EIF4EBP1* or mouse *Acaca* were synthesized and custom cloned (Genewiz) into BsmBI restriction site of gRNA-dCas9-KRAB GFP (gift from Charles Gersbach - [Addgene plasmid #71237 [http://n2t.net/addgene:71237]; RRID:Addgene_71237). sgRNA sequences can be found in Supplementary Table 2.

**cDNA expression plasmids.** The cDNA sequences of human 4EBP1 (T37A/T46A) [4EBP1$^{AA}$] and 4EBP1$^{AA}$ (Y54A/L59A) (4EBP1$^{AA, YL}$) were synthesized and custom cloned (Genewiz) into the EcoRI restriction site of the pLJM1 expression vector (gift from Joshua Mendell - Addgene plasmid #91980 [http://n2t.net/addgene:91980] RRID:Addgene_91980). The cDNA sequence of human *ACACA* flanked by three HA tag sequences in 3' was synthesized and assembled (Vector Builder) in a custom made bacterial vector containing a human ubiquitin C promoter, referred as pUb-ACC1-HA. The 5'UTR of human *ACACA* isoform 3 was synthesized and custom cloned (Genewiz) into the SacI restriction site of pUb-ACC1-HA vector. To generate pCDNA3.1-ACC1-HA plasmid, human *ACACA* cDNA flanked by three HA tag sequences was custom subcloned (Genewiz) from pUb-ACC1-HA into pCDNA3.1 using the NotI and XbaI restriction sites.

## siRNA transfections

Cells were transfected at ~25% confluency in 6-well plates with 25 nM control ON TARGET plus non-targeting siRNA (Dharmacon) or with 25 nM of single siRNAs targeting human and mouse *EIF4E*, human or mouse *FASN*, human or mouse *ACACA*, and mouse *Eif4ebp1* and *Eif4ebp2* using siLentFect transfection reagent (Bio-Rad) according to the manufacturer's instructions. When indicated, cells were glucose starved 48 h post-transfection. siRNA sequences can be found in supplementary table 1.

## Virus production and viral transduction of cell lines

HEK293-T cells were transfected with expression vectors and lentiviral packaging plasmids psPAX2 (gift from Didier Trono - Addgene plasmid #12260 [http://n2t.net/addgene:12260] RRID:Addgene_12260) and pMD2.G (Didier Trono - Addgene plasmid #12259 [http://n2t.net/addgene:12259]; RRID:Addgene_12259) in a ratio of 4:3:1 using CalFectin transfection reagent (Signagen) according to the manufacturer's guidelines. Medium was harvested 72 h post-transfection, passed through a 0.45 μm nitrocellulose filter and frozen at −80 °C. Recipient cells were seeded in 6-well plates and were infected the next day when reaching ~50% confluency. For infection, 0.3 ml of virule-containing medium was added to each well in a final volume of 2 mL medium containing 8 μg/ml polybrene. Stable cell lines were either selected with 2 μg/ml puromycin or 1 mg/ml G418, or FACS sorted for cells expressing GFP.

## Immunoblot analyses of protein expression

Cells were lysed in RIPA buffer (150 mM NaCl, 50 mM Tris-HCl, pH 8, 1% Triton X-100, 0.5% sodium deoxycholate, and 0.1% SDS) supplemented with cOmplete™, EDTA-free Protease Inhibitor Cocktail (Sigma) and phosphatase inhibitors (PhosphoSTOP, Roche). Frozen tumor samples were crushed to dust using a mortar and pestle before lysis. Cell lysates were centrifuged at 14,000 x g for 15 min at 4 °C and supernatants were collected. Protein concentration was measured using the Pierce™ BCA Protein Assay Kit (Thermo Fisher Scientific) according to manufacturer's protocol. Protein lysates were resolved by SDS-PAGE and transferred to nitrocellulose membranes (GE Healthcare). Membranes were blocked with 5% BSA TBS-Tween (20 mM Tris-HCl, pH 7.4, 150 mM NaCl, 0.1% Tween 20) and probed with the primary antibodies indicated in the key resources table. Secondary anti-mouse (926-32210, Li-Cor) or anti-rabbit (926-32211, Li-Cor) antibodies were used and fluorescent signal was detected with the LI-COR Odyssey CLx system. Source data are provided as a Source data file.

## Immunohistochemistry (IHC) staining

IHC for 8-hydroxy-2'- deoxyguanosine (8-oxo-dG) was performed on formalin-fixed paraffin-embedded (FFPE) sections using standard protocols. Briefly, sections were deparaffinized and rehydrated before being incubated with proteinase K at 37 °C for 30 min and washed in PBS. DNA was denatured with 2 N HCL for 5 min at room temperature and then neutralized in 5 M Tris base for 5 min at room temperature. Tissue sections were then incubated with primary antibody (1:250) for 1 h. Tissue sections with bound primary antibody were then incubated with the appropriate secondary antibody (1:500), followed by Chromomap DAB detection.

## RNA analysis

RNA was extracted using the RNeasy mini kit (QIAgen) according to manufacturer's instructions. cDNAs were synthesized from total RNAs using either QuantiTect Reverse Transcription Kit (QIAgen) or High-Capacity cDNA Reverse Transcription Kit (Applied Biosystems) according to manufacturer's instruction. The cDNAs were quantified by real-time PCR analysis using SYBR Green Master Mix (Bio-Rad). The primer sequences are listed in supplementary table 3. As internal controls, L32 or PPIA, GusB and β-actin were amplified.

## ROS measurements

Cells were incubated with 5 μM chloromethyl-2',7'-dichlorodihydrofluorescein diacetate (CM-H2DCFDA) at 37 °C for 20 min. Cells were harvested and resuspended in PBS. Green fluorescence intensity was measured with a CytoFLEX flow cytometer (Beckmann Coulter). Data analysis was performed with FlowJo 10 software (FlowJo).

## Reduced and oxidized glutathione measurements

Cells were seeded into 12-well plates and allowed to attach overnight. Cells were collected and cellular concentrations of reduced and total GSH were quantified using the GSH-Glo assay kit, according to the manufacturer's protocol (Promega). Luminescence was measured using the Spark® plate reader (Tecan).

## Liquid chromatography–mass spectrometry analysis

To measure metabolites, metabolite extraction was performed, in a mixture ice/dry ice, by a cold two-phase methanol–water–chloroform extraction[66,67]. The samples were resuspended in 700 μl of precooled methanol/water (5/3) (v/v) and 100 μL of 13 C yeast internal standard. Afterwards, 500 μl of precooled chloroform was added to each sample. Samples were vortexed for 10 min at 4 °C and then centrifuged (max. speed, 10 min, 4 °C). The methanol–water phase containing polar metabolites was separated and dried using a vacuum concentrator at 4 °C overnight and stored at −80 °C. The measurement of metabolites was performed by liquid chromatography using high resolution and/or triple quadrupole mass spectrometer. For high resolution mass spectrometry, a Dionex UltiMate 3000 LC System coupled to a Q Exactive Orbitrap mass spectrometer (Thermo Scientific) operating in negative mode was used. Metabolite separation was performed at 25 °C with an Ultra High Performance Liquid Chromatography (UHPLC) from Thermo Scientific on a HILIC Fusion (P) column (150 × 2.1 mm, 5 μm). Data was collected and integrated using Xcalibur software (Thermo Scientific). Alternatively, targeted measurements of polar metabolites were performed with a 1290 Infinity II HPLC (Agilent) coupled to a 6470 triple quadrupole mass spectrometer (Agilent). Samples were injected onto a iHILIC-Fusion(P) column. Data analysis was performed with the Agilent software Masshunter. Metabolite levels were normalized to a fully 13C-labelled yeast extract and protein content.

## NADP$^+$/NADPH and NAD$^+$/NADH measurements

NADP/NADPH-Glo™ kit (Promega) and NAD/NADH-Glo™ kit (Promega) were also used to measure NADP$^+$ and NADPH, and NAD$^+$ and NADH, respectively. Cells were lysed in a base solution (100 mM sodium carbonate, 20 mM sodium bicarbonate, 10 mM nicotinamide, 0.05% Triton X-100) containing 1% of Dodecyltrimethylammoniumbromid (DTAB). Cell lysates were split in two equal fractions. The pH of one of the fraction was adjusted by adding 0.4 N HCl according to the manufacturer's protocol. Both fractions were then heated for 15 min at

60 °C and subsequently incubated at RT for 10 min. According to the manufacturer's protocol, before adding the detection reagent, Trizma base or HCl/Trizma solution were used to adjust pH of each fraction. Finally, luminescence of each fraction was analyzed with Spark® plate reader (Tecan) to measure NADP+ and NADPH or NAD+ and NADH levels, and the NADP+/NADPH ratio or NAD+/NADH ratio was calculated.

## Protein synthesis rate

To quantify levels of newly synthesized proteins, 50 µM of azidohomoalanine (AHA) (Thermo Fisher Scientific, Massachusetts) was added to the cell culture medium and cells were incubated for 4 h. Cells were then washed with ice-cold PBS, collected and lysed with EDTA-free RIPA lysis buffer (150 mM NaCl, 50 mM Tris pH 8, 1% Triton X-100, 0.5% sodium deoxycholate, 0.1% SDS). The concentration of proteins was measured by bicinchoninic acid assay using Pierce™ BCA Protein Assay Kit (Thermo Fisher Scientific), and a Click reaction was performed with Click-iT® Protein Reaction Buffer Kit (Thermo Fisher Scientific) according to manufacturer's instructions.

## Cell proliferation

To assess cell proliferation, cells plated in 6-well plates were incubated in fresh medium containing 10 µM 5-ethynyl-2'-deoxyuridine (EdU) (Invitrogen) for 60 min at 37 °C. EdU staining was conducted using Click-iT™ EdU Alexa Fluor™ 488 Flow Cytometry Assay Kit (Invitrogen) according to the manufacturer's protocol. Briefly, cells were harvested, fixed with 4% paraformaldehyde in phosphate buffer saline (PBS) for 15 min, and permeabilized with 1X Click-iT™ saponin-based permeabilization reagent. Cells were incubated with a Click-iT™ reaction cocktail containing Click-iT™ reaction buffer, CuSO₄, Alexa Fluor® 488 Azide, and reaction buffer additive for 30 min while protected from light. Green fluorescence intensity was measured with a CytoFLEX flow cytometer (Beckmann Coulter). Data analysis was performed with FlowJo 10 software (FlowJo).

## Cell death assays

Cell death was measured by flow cytometry using propidium iodide (PI) staining. Briefly, attached and detached cells were harvested, centrifuged and resuspended in PBS containing 1 µg/ml PI (Sigma). Cell death quantification was performed using a CytoFLEX flow cytometer (Beckmann Coulter). A minimum of 50,000 events were recorded for each replicate. All samples were run following the gating strategy in Supplementary Fig. 9. Data analysis was performed with FlowJo 10 software (FlowJo).

## Measurement of mitochondrial activity and mitochondrial mass

Mitochondrial membrane potential was measured using TMRE (tetramethylrhodamine, ethyl ester) mitochondrial membrane potential assay kit (Cayman Chemical, Ann Arbor, MI, USA) and mitochondrial mass was measured using MitoTracker™ Green FM (Thermo Fisher Scientific). Cells were plated in 6-well plates and grown in complete medium or glucose starved for 24 h. TMRE was added to the media to a final concentration of 500 nM and incubated for 20 min at 37 °C. Cells were washed with PBS and harvested using 200 µL of trypsin. Cells were collected with 800 µL of 0.2% BSA containing PBS and fluorescence was measured using flow cytometry.

## Soft agar colony assays

Cells were plated in 6-well plates with 8000 cells per well in DMEM 10% FBS or DMEM 10% bovine calf serum in a top layer of 0.25% agar added over a base layer of 0.4% agar in DMEM 10% FBS or DMEM 10% bovine calf serum. Cells were fed once a week with 1 ml of corresponding medium onto the top layer. Where indicated, NAC (5 mM), Catalase (200 U/ml), TROLOX (100 µM), or TOFA (10 µM) were added to the top

agar layer, as well as twice per week in the feeder medium. After 2–3 weeks at 37 °C, colonies were stained with 0.01% crystal violet and 10 random fields were counted manually for each well. The percentage of colony forming cells was calculated.

## ¹⁴C labeling and fatty acid synthesis activity

Cells were glucose starved for 24 h and labeled with 10 µCi of [1-¹⁴C]-acetate (Perkin Elmer) in the last 18 h. Cells were snapped frozen and lipids were extracted by methanol-water-chloroform extraction. Phase separation was achieved by centrifugation at 4 °C. Radioactivity in the chloroform phase containing fatty acids was quantified by liquid scintillation counting and values were normalized to protein concentration determined in the dried protein interphase.

## ³H labeling and fatty acid oxidation activity

100,000 cells were incubated in complete growth medium in 12-well plates (1 ml per well) for 24 h. Cells were glucose starved for 24, 16, 6, or 0 hours. For the last 6 hours of the experiment, cells were treated with fat-free and serum-free media supplemented with 100 µM palmitic acid and 2 µCi/ml [9,10-³H(N)] palmitic acid (0.5 ml per well). 0.4 ml of medium from each well were transferred to 1.75 ml Eppendorf tubes containing a water-soaked paper filter attached to its cap. The samples were incubated for 48 h at 37 °C. The paper filter was placed in a scintillation vial together with 100 µl water used to wash the cap. Values were normalized by protein concentration and by the percentage of ³H collected. The amount of palmitic acid that was oxidized was calculated.

## Transient expression of proteins

HEK 293 cells were seeded in 6-well plates to reach 50% confluency on the day of transfection. Cells were transfected with 1 µg of pCDNA3.1 or with 500 ng of pCDNA3.1-ACC1-HA completed with 500 ng of pCDNA3.1 using CalFectin transfection reagent (Signagen) according to the manufacturer's guidelines. Cells were harvested 24 h or 48 h post-transfection for further processing.

## 5'UTR Luciferase assays

The 5'UTR Firefly Luciferase reporter plasmids were custom cloned (Genewiz) by inserting the 5'UTR of human *ACACA* isoform 3 into the SacI and BglII restriction sites of pGL3 control vector (Promega).

For transfection, HEK293 cells were seeded in 12-well plates and transfected with 250 ng of each 5'UTR Firefly Luciferase reporter and 3 ng Renilla Luciferase expressing pRL null plasmid (Promega), completed to 500 ng DNA with pcDNA3.1 plasmid, using CalFectin transfection reagent (Signagen) according to the manufacturer's guidelines. Cells were harvested 48 h post-transfection and activity of Firefly and Renilla Luciferase were sequentially determined using the Dual-Luciferase Reporter Assay System (Promega) and analyzed with the Spark® plate reader (Tecan). All samples were performed in triplicate and the final luciferase quantification was formulated as the ratio of Firefly luciferase to Renilla luciferase luminescence.

## Polysome analysis

Cells were treated with 10 µg/ml of cycloheximide for 10 min, washed twice with PBS containing 100 µg/ml cycloheximide, then cells were scrapped and collected. Cells were pelleted by centrifugation (300 x g, 5 min, 4 °C), lysed with 434 µl of lysis buffer (50 mM Tris-base pH 8, 2.5 mM MgCl₂, 1.5 mM KCl, 115 µg/ml cycloheximide, 2.3 mM DTT and 0.27 U/µl RNaseOUT [Thermo Fisher Scientific]), and vortexed. 25 µl of 100% Triton X-100 and 25 µl of 10% sodium deoxycholate were added to the cell lysates, which were vortexed and centrifuged (17,800 x g, 2 min, 4 °C). 50 µl of the lysates were saved as the total fraction and the remaining were loaded on top of a three layers sucrose gradient (5%, 34% and 55% sucrose) that were prepared by dissolving sucrose in

gradient buffer (4 mM HEPES pH 7.6, 20 mM KCl, 1 mM $MgCl_2$). The lysates loaded on top of the sucrose gradient were subjected to ultracentrifugation (229,884 x g, 2.5 h, 4 °C). The polysome profile was read using a piston gradient collector (Biocomp) fitted with a UV detector (Tirax). Three polysomal fractions were collected and placed in Trizol (Sigma-Aldrich Company, location). RNA was extracted from frozen fractions using manufacturer's instructions.

## Proteomic analysis

For mass-spectrometric analysis, samples were cleaned up by a short SDS gel. Gel pieces were reduced and alkylated, followed by digestion with trypsin. Peptides were extracted with 0.1% trifluoroacetic acid and subjected to liquid chromatography. For peptide separation over a 120 min gradient, an Ultimate 3000 Rapid Separation liquid chromatography system (Thermo Scientific) equipped with an Acclaim PepMap 100 C18 column (75 μm inner diameter, 25 cm length, 2 μm particle size from Thermo Scientific) was used. Mass-spectrometric analysis was carried out on an Obitrap Elite mass spectrometer (Thermo Scientific) operating in positive mode and equipped with a nano electrospray ionization source. Capillary temperature was set to 275 °C and source voltage to 1.4 kV. Survey scans were carried out in the Orbitrap mass analyzer over a mass range from 350 to 1700 m/z at a resolution of 60,000 (at 400 m/z). The target value for the automatic gain control was 3,000,000 and the maximum fill time 200 ms. For fragment analysis the 20 most intense peptide ions (minimal signal intensity 500, excluding singly charged ions) were isolated, transferred to the linear ion trap (LTQ) part of the instrument and fragmented using collision induced dissociation (CID). Peptide fragments were analyzed at a resolution of 5400 (at 400 m/z). Already fragmented ions were excluded for fragmentation for 45 s.

Acquired spectra were searched using Mascot 2.4 within Proteome Discoverer version 1.4 against the UniProt database (human; including isoforms; date 2016-05-29). Carbamidomethyl at cysteines was set as fixed modification and methionine oxidation were considered as variable modifications, as well as tryptic cleavage specificity (cleavage behind K and R) with a maximum of two missed cleavage sites. Predefined values were used for other parameters including a false discovery rate of 1% on peptide level, a main search precursor mass tolerance of 10 ppm and mass tolerance of 10 mmu for fragment spectra. Label-free quantification was performed with Progenesis QI for Proteomics (Version 2.0, Nonlinear Dynamics, Waters Corporation, Newcastle upon Tyne, UK).

## Bioinformatics analyses of gene and protein expression patterns in human tissue samples

For gene expression analysis, RNA-seq data from TCGA and the GTEx projects were analyzed with Gepia[68]. For survival analysis, RNA-seq and microarray data were analyzed with Kaplan-Meier Plotter[69] or by using the Chinese Glioma Genome Atlas (CGGA)[70]. For protein expression analysis of 4EBP1 level, CPTAC GBM proteomic data were analyzed with GraphPad Prism. For co-expression analysis of 4EBP1 and ACC1 levels, CPTAC and TCGA GBM proteomic data were analyzed by cBioportal[71,72]. For details of the cohorts, see the key resources table.

## Quantification and Statistical Analysis

All experiments were, if not otherwise stated, independently carried out at least three times. Statistical significance was calculated using Student's t-test in GraphPad Prism 8. The data are represented as means +/− standard deviation. A p-value of less than 0.05 was considered to be significant.

## Reporting summary

Further information on research design is available in the Nature Portfolio Reporting Summary linked to this article.

## Data availability

The mass spectrometry proteomics data have been deposited to the ProteomeXchange Consortium via the PRIDE partner repository with the dataset identifier PXD040931. All other datasets generated and analyzed in this study are provided within the manuscript and the accompanying Supplementary Figs. or from the corresponding authors upon reasonable request. Source data are provided with this paper.

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

## Acknowledgements

We would like to thank Maya Bar and Björn Stork for helpful discussions. We would like to thank Marc B. Hershenson (University of Michigan) for providing us the MSCV puro-4EBP1 (T37A/T46A) plasmid. This work was supported by funds from the Israel Cancer Association (grant no. 20220143), the ISRAEL SCIENCE FOUNDATION (grant no. 1436/19) and the NIBN to B.R.; the Dr. Rolf M. Schwiete Stiftung (grant no. 2020-018) to L.H. and G.L.; the Deutsche Forschungsgemeinschaft (grant no. LE 3751/2-1), the German Cancer Aid (grants no. 70112624 and 70115129), the Elterninitiative Düsseldorf e.V. (grant no. 701910003), and the Research Commission of the Medical Faculty, Heinrich Heine University Düsseldorf (grants no. 2016-056 and 2020-044) to G.L.; the Matthias-Lackas foundation, the Dr. Leopold und Carmen Ellinger foundation, the European Research Council (ERC CoG 2023 #101122595), Dr. Rolf M. Schwiete foundation (2021-007, 2022-31), the Deutsche Forschungsgemeinschaft (DFG 458891500), the German Cancer Aid (DKH-7011411, DKH-70114278, DKH-70115315, DKH-70115914), the SMARCB1 association, the Ministry of Education and Research (BMBF; SMART-CARE and HEROES-AYA), the Fight Kids Cancer foundation (FKC-NEW-targets), the KiTZ-Foundation in memory of Kirstin Diehl, the KiTZ-PMC twinning program, the German Cancer Consortium (DKTK, PRedictAHR), and the Barbara and Wilfried Mohr foundation to T.G.P.G.; the German Cancer Aid (DKH-70114131) and the German Academic Scholarship Foundation to C.M.F.; the European Research Council under the ERC Consolidator Grant Agreement n. 771486–MetaRegulation, FWO—Research Projects (G098120N and G0B4122N), KU Leuven—FTBO (ENN-OZFTBO-P7015), King Baudouin Foundation (2021-J1990112-22260), Beug Foundation (Metastasis Prize 2021) and Fonds Baillet Latour (Grant for Biomedical Research 2020) to S.-M.F. The results published here are in part based on data generated by TCGA Research Network (https://www.cancer.gov/tcga).

## Author contributions

Conceptualization, T.L., K.A., K.V., L.H., B.R., and G.L.; Methodology, T.L., K.A., K.V., L.H., K.S., B.R., and G.L.; Investigation, T.L., K.V., L.H., K.A., Z.B., M.S., R.M., K.S., M.P., K.Vr., S.C., C.M.F., C.H., A.K., B.H., K.B., D.P., A.S., B.R., and G.L.; Resources, A.V-T., U.K., and M.E.; Writing—Original Draft, B.R. and G.L.; Writing—Review & Editing, K.V., L.H., K.A., T.L., S-M.F., B.R., J.K.M.L., and G.L.; Visualization, L.H., K.V., B.R., and G.L.; Supervision, K.St., M.R., T.G.P.G., A.S.R., S-M.F., A.Sc., G.R., B.R. and G.L.; Funding Acquisition, B.R., J.K.M.L. and G.L.

## Funding

## Competing interests

B.R. and G.L. filed for a patent based on these findings. The remaining authors declare no competing interests.

## Additional information

¹Department of Life Sciences, Faculty of Natural Sciences, Ben-Gurion University of the Negev, Beer-Sheva 84105, Israel. ²The National Institute for Biotechnology in the Negev, Ben-Gurion University of the Negev, Beer-Sheva 84105, Israel. ³Institute of Neuropathology, University Hospital Düsseldorf and Medical Faculty, Heinrich Heine University, 40225 Düsseldorf, Germany. ⁴Biochemistry and Molecular Biology, Theodor-Boveri-Institute, 97074 Würzburg, Germany. ⁵Division of Tumor Metabolism and Microenvironment, German Cancer Research Center (DKFZ), 69120 Heidelberg, Germany. ⁶Department of Pediatric Oncology, Hematology, and Clinical Immunology, University Hospital Düsseldorf and Medical Faculty, Heinrich Heine University, 40225 Düsseldorf, Germany. ⁷Laboratory of Cellular Metabolism and Metabolic Regulation, VIB-KU Leuven Center for Cancer Biology, VIB, 3000 Leuven, Belgium. ⁸Laboratory of Cellular Metabolism and Metabolic Regulation, Department of Oncology, KU Leuven and Leuven Cancer Institute (LKI), 3000 Leuven, Belgium. ⁹Division of Translational Pediatric Sarcoma Research, German Cancer Research Center (DKFZ), German Cancer Consortium (DKTK), 69120 Heidelberg, Germany. ¹⁰Hopp Children's Cancer Center (KiTZ), 69120 Heidelberg, Germany. ¹¹German cancer consortium (DKTK) partner site Essen/Düsseldorf, 40225 Düsseldorf, Germany. ¹²Molecular Proteomics Laboratory, Biomedical Research Center (BMFZ), Heinrich Heine University, Medical Faculty, Düsseldorf, Germany. ¹³Institute of Biochemistry and Molecular Biology I, Medical Faculty, Heinrich Heine University, 40225 Düsseldorf, Germany. ¹⁴Clinic for

Neurosurgery, University Hospital Düsseldorf and Medical Faculty, Heinrich Heine University, 40225 Düsseldorf, Germany. [15]Experimental and Clinical Research Center, Max-Delbrück Center for Molecular Medicine and Charité—Universitätsmedizin Berlin, corporate member of Freie Universität Berlin and Humboldt-Universität zu Berlin, 13125 Berlin, Germany. [16]Charité - Universitätsmedizin Berlin, Corporate Member of Freie Universität Berlin and Humboldt-Universität zu Berlin, Department of Radiation Oncology, 13353 Berlin, Germany. [17]Molecular and Experimental Surgery, University Clinic for General-, Visceral, Vascular- and Transplantation Surgery, Faculty of Medicine and University Medicine, Otto-von-Guericke-University, 39120 Magdeburg, Germany. [18]The Shraga Segal Department of Microbiology, Immunology and Genetics, Faculty of Health Science, Ben-Gurion University of the Negev, Beer-Sheva 84105, Israel. [19]Faculty of Health Sciences, Ben-Gurion University of the Negev, Beer-Sheva 84105, Israel. [20]Institute of Pathology, Heidelberg University Hospital, 69120 Heidelberg, Germany. [21]These authors contributed equally: Tal Levy, Kai Voeltzke, Laura Hruby, Khawla Alasad. [22]These authors jointly supervised this work: Barak Rotblat, Gabriel Leprivier. ✉e-mail: rotblat@bgu.ac.il; gabriel.leprivier@med.uni-duesseldorf.de

