## [Peer Review File · Nature Communications]

mTORC1 regulates cell survival under glucose starvation through 4EBP1/2-mediated translational reprogramming of fatty acid metabolismREVIEWER COMMENTS

Reviewer #1 (Remarks to the Author):

Downstream of mTORC1, 4EBP1/2 function as inhibitors of mRNA translation initiation. Levy and colleagues discovered that the mTORC1 substrate 4EBP1/2 plays a pro-survival role under conditions of energetic stress. The authors show that, in the absence of glucose, 4EBP1/2 selectively inhibits the translation of ACC1, thereby preventing NADPH-consuming fatty acid synthesis and promoting redox balance. Additionally, the authors suggest a critical role for the 4EBP1-ACC1 axis in the context of glioma or oncogene-transformed cells.

While the study is interesting, several concerns are presented below:

1. Fig. 2D. The metabolite heatmap indicates that NADH levels are elevated during glucose deprivation, which is unexpected, as glucose deprivation usually results in a cessation of NADH production. Is this due to the prolonged period of glucose starvation (24 hours)? A time-dependent study would help to clarify this observation.
2. Fig. 3A. It is surprising that, of all the proteins affected by 4EBP1 during glucose starvation, ACC1 plays such a significant role in redox balance. Does ACC1 overexpression impact NADP/NADPH or NAD/NADH metabolism or lead to cell death, in response to glucose deprivation? Similarly in Fig, 4H. Does the cell death phenotype rely on alterations in the NADPH/NADP ratio linked to ACC1 function?
3. 5E and 5I. The impact of 4E and ACC1 loss on tumor growth in the NT2197 context is remarkable. However, it is unclear whether these effects are due to alterations in redox or possibly a secondary role of ACC1 (moonlighting?). Is ACC1 overexpression capable of reducing tumor growth in this model? If so, is the enzymatic function of ACC1 required for this effect?

Minor:

1. Fig. 4C. The qPCR data can be repeated to improve data quality and interpretation.
2. Fig. 4A. Although the authors' conclusion is that ACLY is not influenced in the 4E knockout, it is apparent that ACLY actually undergoes a notable increase that is similar to ACC1. Quantification of the blots would clarify this point.

Reviewer #2 (Remarks to the Author):

Levy, Voeltzke, and colleagues provide valuable insights into how mTORC1 signaling supports cell growth when cells are subjected to glucose restriction. The authors demonstrate that 4EBP1/2, downstream targets and effectors of mTORC1, play a protective role against cell death in mammalian and yeast cells

during glucose starvation. The 4EBP1/2 proteins suppress the translation of ACC mRNA transcript, a crucial enzyme in converting acetyl-CoA to malonyl-CoA for fatty acid synthesis and NADPH consumption, particularly by fatty acid synthase. During glucose starvation, cell viability is sustained by 4EBP1/2 through NADPH-dependent inhibition of ROS. These findings have significant implications for cancer research as the authors have uncovered that oncogene-transformed and glioma cells rely on 4EBP1/2 to control ACC1 expression and redox balance, which lessens the energetic stress caused by glucose starvation. This, in turn, supports the transformation and tumorigenicity of cancer cells in vitro and in vivo. Furthermore, the authors report that high expression of 4EBP1 is linked to poor overall survival rates in cancer patients, supporting a pro-tumorigenic role of 4EBP in cancers.

Overall, this is a very interesting study. However, the authors could further strengthen their research by addressing the limitations mentioned below.

The increase in both NAD(H) and NADP(H) when 4EBP/12 is knocked out suggests that other mechanisms exist, promoting the synthesis of NAD, the precursor of NADP(H). Therefore, the authors should assess the activity of the NAD salvage pathway or de novo NAD synthesis when 4EBP1/2 expression is suppressed.

Additionally, the authors should express a 4EBP cDNA in 4EBP KO cells to demonstrate that the observed effects are not due to clonality resulting from the selection.

The authors observed an increase in NADPH-dependent RRM1 in sh4EBP1/2 cells compared to shScr. The authors should assess whether RRM1 inhibition can also rescue cell death, similarly to ACC knockdown or FASN inhibition under glucose starvation.

Reviewer #3 (Remarks to the Author):

In the manuscript by Levy et al. the authors investigate how mTORC1 inhibition supports cell viability during glucose starvation. Their data suggest that 4EBP1/2, activated upon mTORC1 inhibition, represses ACC1, blocking fatty-acid synthesis and preventing NADPH-consumption. Thus, activated 4EBP1/2 mitigates oxidative stress. The authors also suggest that transformed cells and glioma cells utilize this mechanism to reduce oxidative stress. The study is of interest and elucidates a potentially important role for 4EBP1 under glucose starved conditions.

Figure 1I. There appears to be decreased colony formation with eap1 suggesting even under control conditions the mutant has an effect. This should be acknowledged and/or quantified.

-The statement below should be more specific as there are other important functions of 4EBP1 that may contribute to tumorigenesis unless the authors have evidence to the contrary.

Statement” Thus, we conclude that 4EBP1/2 support oncogenic transformation by negatively regulating ACC1 and oxidative stress”

Figure 6.

A. The panel comparing Grade 2 and 3 diffuse glioma (comprised of largely IDH-mutant diffuse astrocytoma and oligodendroglioma) with G4 GBM (>94% IDH-wildtype) is misleading and does not reflect our current knowledge of these diseases. The expression of many genes are different but this cannot be interpreted as a progression across grades as the diseases are different.

B. Can the authors define “glioma”? Are these data inclusive of G2-4 or G2-3? What is the expression levels in astrocytoma vs. oligodendroglioma? If expression levels are different in the two types of IDH-mutant glioma then the curve could largely reflect their OS. The analysis would be best done comparing within a molecular subtype if possible.

G-J. While these data show a role for 4EBP1 in human and murine glioma this has been shown previously and the importance of 4EBP1 in human GBM was the focus of a recent study (Fan et al. PMID: 28292440). Can the investigators rescue the defect or show it is related to dysregulated fatty acid synthesis? The difference in OS is also not dramatic. Given recent studies this is somewhat surprising. Can the authors comment? Is knockdown maintained?

H. Here the authors use dox to induce knockdown. Yet in Panels I-K the constructs are on constitutively. Why is this set-up different and why is an inducible system not required in the brain?

-The authors examine total eIF4EBP expression in tumors and glioma. Studies suggest that phosphorylation of 4EBP1 (inactivation) is common in GBM and blocking mTORC1 suppresses glioma growth in vivo. How do the authors reconcile this with their findings?

Reviewer #4 (Remarks to the Author):

This manuscript reports the role of eukaryotic initiation factor 4E binding proteins 1/2 (4EBP1/2) in cell survival under glucose starvation conditions. Activated 4EBP1/2 upon mTORC1 complex inhibition protects cells from glucose starvation. Mechanistically, 4EBP1/2 activation inhibits ACC1 translation, leading to reduced fatty acid synthesis, increased NADPH, and reduced oxidative stress when glucose is low. Knockout of 4EBP1/2 inhibits tumor growth and additional knockdown of ACC1 rescued this

phenotype. Overexpression of 4EBP1/2 promotes Hela tumor growth in vivo. I have the following criticisms:

1. Overall, it is not clear how to associate the effect of 4EBP1/2 on cell survival under glucose starvation in vitro with the role of 4EBP1/2 in tumor growth in vivo.
2. Cells (including tumor cells) in the body are rarely under complete glucose starvation conditions. Tissue culture medium usually contains 4.5 g/L glucose. Does 4EBP1/2 have a similar effect on cell survival in low glucose (e.g., 1g/L) culture medium?
3. Figures 1E, 1G, 5, and 6: does the 4EBP substrate eIF4E mediate the role of 4EBP1/2?
4. AMPK regulates ACC1 activity under low glucose or glucose starvation conditions. Does AMPK signaling play a role in the effect of 4EBP1/2 on cell survival and tumor growth?
5. Figure 5E: 4EBP1 rescue should be included as a control.
6. Figure 5G: the eIF4E-non-binding mutant, 4EBP1AA;YL, should be included as a control, and overexpression of ACC1 should be included to strengthen the conclusion.
7. Figures 5 and 6: what are the glucose levels in tumors?

REVIEWER COMMENTS

Reviewer #1 (Remarks to the Author):

Downstream of mTORC1, 4EBP1/2 function as inhibitors of mRNA translation initiation. Levy and colleagues discovered that the mTORC1 substrate 4EBP1/2 plays a pro-survival role under conditions of energetic stress. The authors show that, in the absence of glucose, 4EBP1/2 selectively inhibits the translation of ACC1, thereby preventing NADPH-consuming fatty acid synthesis and promoting redox balance. Additionally, the authors suggest a critical role for the 4EBP1-ACC1 axis in the context of glioma or oncogene-transformed cells.

While the study is interesting, several concerns are presented below:

1. Fig. 2D. The metabolite heatmap indicates that NADH levels are elevated during glucose deprivation, which is unexpected, as glucose deprivation usually results in a cessation of NADH production. Is this due to the prolonged period of glucose starvation (24 hours)? A time-dependent study would help to clarify this observation.

We would like to thank the reviewer for pointing this out. We indeed observed an increase of NADH levels upon 24h glucose starvation. It is worth noting that this was accompanied by a rise of NAD⁺ of greater magnitude, resulting in a net reduction of the NADH/NAD⁺ ratio upon 24h glucose starvation (Figure 2E). These data together suggest that NAD⁺ biosynthesis is stimulated by 24h glucose starvation. This is supported by a previous report showing that glucose starvation promotes the NAD⁺ salvage pathway by inducing transcription of the enzyme Nampt, increasing the NAD⁺/NADH ratio (PMID: 18477450). To further characterize the impact of glucose starvation on NADH and NAD⁺ levels at shorter time points, we measured NADH and NAD⁺ at 0h, 1h and 6h of glucose starvation in MEFs WT and DKO, as well as in 293 shScr and sh4EBP1/2 cells (new data in figure S2I-L and new text on p10, first paragraph). While NADH and NAD⁺ levels are unaffected at 1h of glucose starvation, these are substantially increased at 6h of glucose starvation. Together with the above-mentioned previous study, these support that NAD⁺ biosynthesis is induced past 1h of glucose starvation.

2. Fig. 3A. It is surprising that, of all the proteins affected by 4EBP1 during glucose starvation, ACC1 plays such a significant role in redox balance. Does ACC1 overexpression impact NADP/NADPH or NAD/NADH metabolism or lead to cell death, in response to glucose deprivation?

ACC1 is important in regulating the intracellular redox balance as it is an integral component of fatty acid synthesis, the most NADPH-consuming process in a cell (PMID: 24805240). More specifically, it was reported that overexpression of ACC1 induced an increase of H₂O₂ levels under matrix detachment, a stress condition with similar metabolic consequences as glucose starvation (PMID: 22660331). In addition, overexpression of ACC2, a close paralogue of ACC1, induced cell death under glucose starvation (PMID: 22660331). To further confirm the role of ACC1 in the control of the redox balance in our system, we overexpressed ACC1 in 293 WT cells and measured NADPH/NADP⁺ ratio (new data in figure S5C and new text p15, third paragraph) as well as the rates of cell death under glucose starvation (new data in figure S5D and new text p15, third paragraph). We found that ACC1 overexpression severely reduced NADPH/NADP⁺ ratio upon glucose starvation, which was accompanied by a higher rate of cell death under these

conditions. This supports that beyond 4EBP1/2, ACC1 has on its own a major impact on the NADPH/ NADP⁺ ratio and the cellular sensitivity towards glucose starvation.

Similarly in Fig, 4H. Does the cell death phenotype rely on alterations in the NADPH/NADP ratio linked to ACC1 function?

We agree with the reviewer that this is an important point. As supportive evidence, in the initial manuscript, we reported that inhibition of ACC by TOFA reduces cell death under glucose starvation (Figure 3C&D), while leading to an increase of NADPH/NADP⁺ ratio in two different 4EBP1/2 deficient cells (Figure 3J&K). To recapitulate these findings upon ACC1 knockdown in 4EBP1/2 knockout MEFs (related to data of figure 4H), we measured the impact of ACC1 knockdown on NADPH/NADP⁺ ratio under glucose starvation (new data in figure S5B and new text p15, third paragraph). We found that knockdown of ACC1, with two different siRNAs, is sufficient to enhance NADPH/NADP⁺ ratio under glucose starvation in 4EBP1/2 knockout cells, supporting that ACC1 inhibition by 4EBP1/2 promotes cell survival by increasing the NADPH/NADP⁺ ratio.

3. 5E and 5I. The impact of 4E and ACC1 loss on tumor growth in the NT2197 context is remarkable. However, it is unclear whether these effects are due to alterations in redox or possibly a secondary role of ACC1 (moonlighting?). Is ACC1 overexpression capable of reducing tumor growth in this model? If so, is the enzymatic function of ACC1 required for this effect?

We would like to thank the reviewer for raising these questions. Our in vitro data support that the impact of 4EBP1/2 loss on the tumorigenicity of NT2197 cells is related to changes in the redox balance as antioxidants treatment rescued the observed phenotype (Figure 5C). We directly addressed the question of what role ACC1 plays in the tumorigenic phenotype driven by 4EBP1/2 in NT2197 cells by performing a rescue experiment. Knockdown of ACC1 in 4EBP1/2 KO NT2197 cells was sufficient to increase tumorigenicity in vivo as compared to control shGFP cells (Figure 5I&J). In addition, we now found using dityrosine immunoblotting or 8-Oxo-2'-deoxyguanosine (8-oxo-dG) IHC that 4EBP1 KO or KD increased oxidative stress in tumors (new data in figure S6F&G, new text p17, first paragraph and in figure S8C&D, new text p19, first paragraph) and that ACC1 KD rescued such increased oxidative stress in 4EBP1/2 KO NT2197 tumors in vivo (new data in figure 6F&G and new text p17, first paragraph), together supporting that the observed phenotypes upon 4EBP1 and ACC1 loss are related to changes in the redox balance.

We understand that the reviewer is curious whether ACC1 overexpression will reduce tumorigenicity in vivo in this specific model and whether ACC1 enzymatic activity contributes to its potential anti-tumorigenic functions in vivo. This question was addressed in a previous study using MCF7 cells (PMID: 22660331), where the authors showed that a constitutively active ACC1 reduces tumorigenicity in vivo.

Due to the large uncertainties linked to the situation in Israel, we are currently not able to perform in vivo experiments.

Minor:

1. Fig. 4C. The qPCR data can be repeated to improve data quality and interpretation.

We repeated the experiment and now show the new data (Figure 4C). While there is higher *ACACA* expression in 293 sh4EBP1/2 versus shScr cells in resting conditions, this difference does not exist during glucose starvation. These findings align with our previous data and show that different mRNA levels do not explain the difference in ACC1 protein levels found in the 4EBP1/2 deficient versus control cells during glucose starvation.

2. Fig. 4A. Although the authors' conclusion is that ACLY is not influenced in the 4E knockout, it is apparent that ACLY actually undergoes a notable increase that is similar to ACC1.

Quantification of the blots would clarify this point.

As suggested by the reviewer, we quantified the blots (now added to figure 4A&B, new text p14, first paragraph) and found that while upon glucose starvation, ACC1 is higher in 4EBP1/2 deficient vs control in both MEFs and HEK293 cells, ACLY is higher only in MEFs. Therefore, ACC1 expression correlates better with the phenotype.

Reviewer #2 (Remarks to the Author):

Levy, Voeltzke, and colleagues provide valuable insights into how mTORC1 signaling supports cell growth when cells are subjected to glucose restriction. The authors demonstrate that 4EBP1/2, downstream targets and effectors of mTORC1, play a protective role against cell death in mammalian and yeast cells during glucose starvation. The 4EBP1/2 proteins suppress the translation of ACC mRNA transcript, a crucial enzyme in converting acetyl-CoA to malonyl-CoA for fatty acid synthesis and NADPH consumption, particularly by fatty acid synthase. During glucose starvation, cell viability is sustained by 4EBP1/2 through NADPH-dependent inhibition of ROS. These findings have significant implications for cancer research as the authors have uncovered that oncogene-transformed and glioma cells rely on 4EBP1/2 to control ACC1 expression and redox balance, which lessens the energetic stress caused by glucose starvation. This, in turn, supports the transformation and tumorigenicity of cancer cells in vitro and in vivo. Furthermore, the authors report that high expression of 4EBP1 is linked to poor overall survival rates in cancer patients, supporting a pro-tumorigenic role of 4EBP in cancers.

Overall, this is a very interesting study. However, the authors could further strengthen their research by addressing the limitations mentioned below.

The increase in both NAD(H) and NADP(H) when 4EBP1/2 is knocked out suggests that other mechanisms exist, promoting the synthesis of NAD, the precursor of NADP(H). Therefore, the authors should assess the activity of the NAD salvage pathway or de novo NAD synthesis when 4EBP1/2 expression is suppressed.

We agree with the reviewer that this point needs to be clarified. When summing up the amount of NAD⁺ to NADH, or the amount of NADP⁺ to NADPH, as a readout for NAD or NADP biosynthesis respectively, we do not observe any significant difference between 293 shScr and sh4EBP1/2 under glucose starvation (data not shown). However, our data showed that NAD(H) and NADP(H) are increased by glucose starvation, but to the same extent in 293 shScr and sh4EBP1/2 cells (Figure 2D). This is in line with a previous report highlighting that glucose starvation promotes the NAD⁺ salvage pathway by inducing transcription of the enzyme Nampt, in turn increasing the

NAD⁺/NADH ratio (PMID: 18477450). To further characterize the impact of glucose starvation on NAD⁺ and NADH levels at shorter time points, we measured NAD⁺ and NADH at 0h, 1h and 6h of glucose starvation in MEFs WT and DKO, as well as in 293 shScr and sh4EBP1/2 cells (new data in figure S2I-L and new text p10, first paragraph). We observed that NAD⁺ and NADH levels are substantially increased at 6h, but not at 1h, of glucose starvation. Furthermore, 4EBP1/2 deficiency does not consistently impact levels of NAD⁺ or NADH at 1h and 6h glucose starvation. Together with the above-mentioned study, our data suggest that glucose starvation promotes NAD⁺ salvage pathway independently of 4EBP1/2 expression.

Additionally, the authors should express a 4EBP cDNA in 4EBP KO cells to demonstrate that the observed effects are not due to clonality resulting from the selection.

We agree with the reviewer that this represents an important control. We have expressed exogenous and constitutively active 4EBP1 (4EBP1-AA) in 4EBP KO MEFs. This was sufficient to strongly reduce cell death under glucose starvation (new data figure S1A and new text p7, first paragraph). In addition, we have expressed the same 4EBP1-AA in NT2197 4EBP1/2 KO cells and injected these cells in the flank of immunocompromised mice. We found that 4EBP1-AA expression led to a significant increase of tumor mass as compared to control NT2197 4EBP1/2 KO cells (new data figure S6D&E and new text p17, first paragraph). In addition, in our initial manuscript we could recapitulate the phenotype observed with 4EBP1/2 KO MEFs (i.e. increased cell death under glucose starvation) and in NT2197 4EBP1/2 KO cells (i.e. decreased tumorigenic potential in vitro and in vivo) in a variety of cell lines using shRNA, CRISPRi or siRNA to deplete 4EBP1 or 4EBP1/2 (Figures 1G, 6D, 6F, 6H&I, S6A, S7G and S8A). Together, these data exclude that the phenotype of 4EBP1/2 KO is due to clonality.

The authors observed an increase in NADPH-dependent RRM1 in sh4EBP1/2 cells compared to shScr. The authors should assess whether RRM1 inhibition can also rescue cell death, similarly to ACC knockdown or FASN inhibition under glucose starvation.

We would like to thank the reviewer for this suggestion. We have inhibited the activity of RRM1 in 293 sh4EBP1/2 using gemcitabine. This treatment only marginally decreased the rates of cell death under glucose starvation (new data in figure S4A and new text p12, first paragraph). Therefore, RRM1 does not appear as a major mediator of the protective function of 4EBP1/2 under glucose starvation, in contrast to ACC1.

Reviewer #3 (Remarks to the Author):

In the manuscript by Levy et al. the authors investigate how mTORC1 inhibition supports cell viability during glucose starvation. Their data suggest that 4EBP1/2, activated upon mTORC1 inhibition, represses ACC1, blocking fatty-acid synthesis and preventing NADPH-consumption. Thus, activated 4EBP1/2 mitigates oxidative stress. The authors also suggest that transformed cells and glioma cells utilize this mechanism to reduce oxidative stress. The study is of interest and elucidates a potentially important role for 4EBP1 under glucose starved conditions.

Figure 11. There appears to be decreased colony formation with eap1 suggesting even under control conditions the mutant has an effect. This should be acknowledged and/or quantified.

We agree with the reviewer that *eap1* deletion has an effect, albeit small, on growth under basal conditions. We changed the text to reflect these findings and it now reads: "While disruption of *eap1* (*eap1Δ*) or *caf20* (*caf20Δ*) had **either a small or** no observable impact, respectively, on the growth capacities in basal, glucose-containing YPD medium, the growth of *eap1Δ* strain, but not of *caf20Δ* strain, was severely compromised in glucose-free YP media, as compared to WT strain (Fig. 1I)" (p8, third paragraph).

-The statement below should be more specific as there are other important functions of 4EBP1 that may contribute to tumorigenesis unless the authors have evidence to the contrary. Statement" Thus, we conclude that 4EBP1/2 support oncogenic transformation by negatively regulating ACC1 and oxidative stress"

We have now changed the initial statement to the following: "Thus, we conclude that 4EBP1/2 support oncogenic transformation **at least in part** by negatively regulating ACC1 and oxidative stress" (p16, third paragraph).

Figure 6.

A. The panel comparing Grade 2 and 3 diffuse glioma (comprised of largely IDH-mutant diffuse astrocytoma and oligodendroglioma) with G4 GBM (>94% IDH-wildtype) is misleading and does not reflect our current knowledge of these diseases. The expression of many genes are different but this cannot be interpreted as a progression across grades as the diseases are different.

We agree with the reviewer. We are changing the label of former Figure 6A (now Figure 6B) to reflect that these are different diseases (new text, p18, second paragraph).

B. Can the authors define "glioma"? Are these data inclusive of G2-4 or G2-3? What is the expression levels in astrocytoma vs. oligodendroglioma? If expression levels are different in the two types of IDH-mutant glioma then the curve could largely reflect their OS. The analysis would be best done comparing within a molecular subtype if possible.

We agree with the reviewer that this point needs to be clarified. In figure 6B, the term "glioma" refers to Grade 2 and Grade 3 diffuse glioma as well as Grade 4 glioblastoma. To ensure of a correlation between *EIF4EBP1* expression and patients' survival in each of these diseases, we generated Kaplan-Meier plots for Grade 2 and Grade 3 diffuse glioma separately. We found no significant association between *EIF4EBP1* expression and survival in each of these gliomas (data not shown), in striking contrast to Grade 4 glioblastoma (former figure 6C, now figure 6A). Therefore, we decided to remove the Kaplan-Meier curves generated for all gliomas (former figure 6B and figure S7E), to avoid any confusion, and put the focus on the Kaplan-Meier curve done with Grade 4 glioblastoma patient data (former figure 6C, now figure 6A).

G-J. While these data show a role for 4EBP1 in human and murine glioma this has been shown previously and the importance of 4EBP1 in human GBM was the focus of a recent study (Fan et al. PMID: 28292440). Can the investigators rescue the defect or show it is related to dysregulated fatty acid synthesis?

We thank the reviewer for this comment and would like to clarify our findings in regard of the above-mentioned publication. Fan *et al.* used different mTOR inhibitors and found a correlation between 4EBP activity (phosphorylation) and the efficacy of mTOR inhibitors. The compounds most efficiently activating 4EBP were best at inhibiting GBM cell growth. This is in line with the known function of 4EBP in blocking cell proliferation (PMID: 20508131). The findings of Fan *et al.* are opposite to our findings as we demonstrated that blocking, instead of activating, 4EBP1 restricts glioma tumorigenic potential and tumor growth in vivo. Together, these suggest that 4EBP1 plays a dual role in GBM depending on the type of stress. On the one hand, 4EBP1 promotes cell survival under glucose-deprived conditions (this manuscript) while, on the other hand, 4EBP1 inhibits cell proliferation under serum or amino acids starvation or upon pharmacological inhibition of mTOR (PMID: 20508131).

We established the link between 4EBP function and fatty acid synthesis using the ACC inhibitor, TOFA, which rescued 4EBP1/2 deficient cells from cell death under glucose starvation (Figure 3C&D) and restored colony formation of 4EBP1/2 deficient cells in soft agar (Figures 5C, 6E, S7I-K). We also used a genetic approach, such that knockdown of the fatty acid synthesis enzymes FASN or ACC1 rescued the survival of 4EBP1/2 deficient cells under glucose starvation (Figures 3E&F, 4H&I), while the knockdown of ACC1 restored the growth of 4EBP1/2 deficient cancer cells in soft agar (Figure 5D), and in vivo (Figures 5I&J, 6J). Taken together, these findings indicate that 4EBP1 protective function and pro-tumorigenic function are undoubtedly linked to the control of fatty acid synthesis.

The difference in OS is also not dramatic. Given recent studies this is somewhat surprising. Can the authors comment? Is knockdown maintained?

We appreciate the point raised by the reviewer and realize that this point needs further clarification. In Fan *et al.* study (PMID: 28292440), the extension of OS is obtained using an mTOR inhibitor, which activates 4EBP1. In our case, we observed an increase OS when employing the opposite strategy which was by blocking 4EBP1 expression. Therefore it is difficult to estimate how dramatic is the difference of OS we observed in comparison to similar models, as there are no reported orthotopic glioma/GBM models in which 4EBP1 was inhibited.

H. Here the authors use dox to induce knockdown. Yet in Panels I-K the constructs are on constitutively. Why is this set-up different and why is an inducible system not required in the brain?

We addressed different questions using inducible vs. stable knockdown. Stable knockdown is used to determine the contribution of 4EBP1 to tumorigenicity. The inducible system is used to determine the role of 4EBP1 in tumor maintenance. Because we found using stable knockdown approach that 4EBP1 promotes tumorigenicity and tumor growth in the flank injection experiments, we thus opted to use the stable system in the orthotopic experiments.

-The authors examine total eIF4EBP expression in tumors and glioma. Studies suggest that phosphorylation of 4EBP1 (inactivation) is common in GBM and blocking mTORC1 suppresses glioma growth in vivo. How do the authors reconcile this with their findings?

We would like to thank the reviewer for pointing to this apparent paradox and agree that this is an important point. While there are studies reporting high amount of 4EBP1 phosphorylation in different tumor types, including gliomas, very few of these studies analyzed the levels of total 4EBP1 protein. As *EIF4EBP1* is overexpressed in numerous tumor entities, this may account for the higher levels of phospho-4EBP1 observed in such tumors. In support to that, by analyzing the correlation between *EIF4EBP1* and phospho-4EBP1 levels in GBM, we found that *EIF4EBP1* mRNA levels are positively correlated with levels of phospho-4EBP1 (new data in figure S8G-I and new text p23, second paragraph). Moreover, it was previously shown that there is a remaining active functional 4EBP1/2 fraction in cases where mTORC1 is active and 4EBP1/2 phosphorylated (PMID: 33177490).

In addition, the levels of 4EBP1 phosphorylation, as well as the activity of mTORC1, is heterogeneous in GBM tumor tissues and is dependent on the distance from blood vessels. It was reported that GBM cells in close proximity to blood vessels exhibit high levels of 4EBP1 phosphorylation and rS6P phosphorylation (PMID: 31056286). In contrast, GBM cells far away from blood vessels show low amounts of 4EBP1 phosphorylation and rS6P phosphorylation (PMID: 31056286). This indicates that 4EBP1 is active in the metabolically challenged areas of GBM tissues.

While it was reported that an mTORC1 inhibitor restricted the growth of orthotopic GBM models in mice (PMID: 28292440), it is known that the effect of mTORC1 inhibitors depends on the vascularization of the tumor. Indeed, it was demonstrated that while mTORC1 inhibition restricts proliferation in vascularized tumor regions, it exhibits the opposite effect in hypovascularized tumor regions by promoting proliferation (PMID: 26144316). The net result is that mTORC1 inhibition promoted tumor progression in a genetically engineered mouse model of pancreatic cancer (PMID: 26144316). Therefore, differences in the levels of metabolic stress between our in vivo models and the orthotopic glioblastoma models used by Fan *et al.* (PMID: 28292440) may explain such seemingly opposite effects (that 4EBP1 inhibition as well as mTORC1 inhibition restrict glioblastoma growth).

Reviewer #4 (Remarks to the Author):

This manuscript reports the role of eukaryotic initiation factor 4E binding proteins 1/2 (4EBP1/2) in cell survival under glucose starvation conditions. Activated 4EBP1/2 upon mTORC1 complex inhibition protects cells from glucose starvation. Mechanistically, 4EBP1/2 activation inhibits ACC1 translation, leading to reduced fatty acid synthesis, increased NADPH, and reduced oxidative stress when glucose is low. Knockout of 4EBP1/2 inhibits tumor growth and additional knockdown of ACC1 rescued this phenotype. Overexpression of 4EBP1/2 promotes Hela tumor growth in vivo. I have the following criticisms:

1. Overall, it is not clear how to associate the effect of 4EBP1/2 on cell survival under glucose starvation in vitro with the role of 4EBP1/2 in tumor growth in vivo.

We agree with the reviewer that this points needs to be clarified. The Brugge's lab previously showed that to survive during matrix detachment, a hallmark of oncogenic transformation, cells must overcome energetic stress due to reduced glucose uptake and increased oxidative stress (PMID: 19693011). In addition, the Hay's lab showed that tumor cells must inhibit fatty acid synthesis to maintain the redox balance as a mean to survive glucose starvation, matrix

detachment and to grow tumors in vivo (PMID: 22660331). It is therefore established that there is a strong link between cell survival during glucose starvation and tumor growth in vivo.

To strengthen the link between the ability of 4EBP1/2 to reduce oxidative stress, which we found to promote survival during glucose starvation, and its tumor-promoting functions, we measured oxidative stress in tumors and found that 4EBP1 KD or 4EBP1/2 KO tumors exhibited increased oxidative stress (as measured by dityrosine immunoblotting or 8-oxo-dG IHC) (new data in figure S6F&G, new text p17, first paragraph and in figure S8C&D, new text p19, first paragraph). This effect was rescued upon ACC1 depletion in NT2197 4EBP1/2 KO tumors (new data in figure S6F&G and new text p17, first paragraph).

2. Cells (including tumor cells) in the body are rarely under complete glucose starvation conditions. Tissue culture medium usually contains 4.5 g/L glucose. Does 4EBP1/2 have a similar effect on cell survival in low glucose (e.g., 1g/L) culture medium?

We would like to thank the reviewer for the suggestion. We measured the rates of cell death of 293 shScr and sh4EBP1/2, WT and 4EBP1/2 DKO MEFs under 2.5 and 5 mM glucose conditions. We found that under both conditions, 4EBP1/2 deficient cells exhibited higher levels of cell death compared to the respective control cells (new data in figure S1J&K and new text p8, first paragraph), similarly to what we initially observed with 1 mM glucose conditions.

3. Figures 1E, 1G, 5, and 6: does the 4EBP substrate eIF4E mediate the role of 4EBP1/2?

We would like to thank the reviewer for raising this question. We understand that it is important to delineate the implication of eIF4E in mediating the protective function and pro-tumorigenic function of 4EBP1/2. In the initial manuscript, we demonstrated that in contrast to 4EBP1-AA, an eIF4E-non-binding mutant, 4EBP1-AA (4EBP1-AA, YL), cannot protect HeLa from glucose starvation (Figure 1E). Regarding the implication of eIF4E in the phenotype observed in different 4EBP1/2 deficient cells under glucose starvation, in the initial manuscript, we provided evidence that knockdown of eIF4E significantly reduced cell death of 4EBP1/2 deficient cells under glucose starvation, namely 4EBP1/2 DKO MEFs and 293 sh4EBP1/2 cells (Figure 1H and Figure S1L, respectively). Together, these data show that eIF4E contributes to 4EBP1/2 protective function under glucose starvation.

To delineate the implication of eIF4E to 4EBP1/2 pro-tumorigenic function in vivo, as for Figure 5, we have overexpressed 4EBP1-AA, YL in HeLa cells and assessed its impact on tumor growth in vivo, in comparison to 4EBP1-AA (new data in Figure 5G-H and new text p17, first paragraph). We found that 4EBP1-AA, YL lost the ability to promote tumor growth compared to 4EBP1-AA. This is in line with a model whereby the 4EBP1/2-mediated inhibition of eIF4E is required for promoting 4EBP1/2 pro-tumorigenic function. To further substantiate this model, in regard of the findings of figures 5 and 6, we knocked down eIF4E in two different 4EBP1/2 deficient cancer cells, namely NT2197 4EBP1/2 KO and in U87 sg4EBP1 cells, and assessed the impact on tumorigenicity in vitro (using soft agar assays). As shown below, while one of the siRNA targeting eIF4E led to a reduction of colony formation, there was no statistically significant difference with the other siRNA as compared to control siRNA (scr). This may be due to different levels of eIF4E knockdown which may have different outputs in such an assay where both proliferation and cell survival are contributing to colony formation. Thus, while a severe knock down of eIF4E is expected to reduce colony formation – as eIF4E is an oncoprotein

stimulating proliferation – a less pronounced reduction of eIF4E expression may promote survival of the colony in anchorage-independent conditions (which represents stress conditions) – similarly to what we showed with survival under glucose starvation (Fig. 1H, S1L). Therefore, it is difficult to further assess the actual implication of eIF4E in the pro-tumorigenic function of 4EBP1/2.

Impact of eIF4E knock down on colony formation of 4EBP1/2 deficient cancer cells. NT2197 4EBP1/2 double KO (DKO) (A) or sg4EBP1#1 U-87 MG (sg6) cells (B) transfected with control (scr) or Eif4e1 targeting siRNAs (si eIF4E#1 and #2) were grown in soft agar for 21 days. Colonies and single cells were counted, and colony formation efficiency was calculated. The level of the indicated proteins was analyzed by immunoblotting. Data are reported as means \pm SD with indicated significance (**p < 0.01).

4. AMPK regulates ACC1 activity under low glucose or glucose starvation conditions. Does AMPK signaling play a role in the effect of 4EBP1/2 on cell survival and tumor growth?

This is an interesting question. AMPK directly regulates ACC1 and mTOR and is activated during energetic stress. While inhibiting ACC1 is essential for AMPK protective function and pro-tumorigenic function, the contribution of mTOR inhibition is not completely understood. Actually, in the article from the Hay's lab (PMID: 22660331) it was shown that inhibiting mTOR, using rapamycin, did not rescue AMPK KO cells from glucose starvation-induced cell death, which suggested that mTOR inhibition is dispensable for AMPK pro-survival function. However, rapamycin efficiently inhibits mTOR-mediated p70S6K phosphorylation but not 4EBP phosphorylation as previously reported (PMID: 19402821). In the initial manuscript, using an mTOR inhibitor that leads to inhibition of 4EBP phosphorylation, i.e. KU-0063794, we found that

mTOR inhibition and consequent 4EBP activation partially rescued AMPK KO cells from cell death under glucose starvation (Figure 1A-C). Furthermore, hyperactive 4EBP1 rescued HeLa cells, which do not activate AMPK during glucose starvation, from glucose starvation-induced cell death and promoted tumorigenicity (Figure 1D&E; Figure 5G&H). Finally, we found that in a variety of cell lines in which AMPK is active under glucose starvation (Figure S2G&H), 4EBP1 promotes survival during glucose starvation (Figure 1F&G). Furthermore, blocking 4EBP1/2 has no impact on AMPK activity under glucose starvation, as we observed by monitoring levels of phospho-AMPK and phospho-ULK1 in 4EBP1/2 deficient versus control cells (Figure S2G&H), indicating that AMPK is not functioning downstream of 4EBP1/2 under these conditions. Together, these findings show that 4EBP1 potentially contributes to AMPK-mediated survival during glucose starvation and tumor growth.

5. Figure 5E: 4EBP1 rescue should be included as a control.

We would like to thank the reviewer for suggesting this experiment. We ectopically expressed 4EBP1-AA in 4EBP1/2 KO NT2197 cells and found that it promoted tumor growth thus, rescuing 4EBP1 KO phenotype in vivo (new data in figure S6D&E and new text p17, first paragraph).

6. Figure 5G: the eIF4E-non-binding mutant, 4EBP1AA;YL, should be included as a control, and overexpression of ACC1 should be included to strengthen the conclusion.

We agree with the reviewer that these are important points. We now added tumor growth data using HeLa cells expressing 4EBP1-AA/YL mutant and found that 4EBP1-AA/YL does not promote tumorigenicity in vivo, in contrast to 4EBP1-AA (new data in Fig. 5G&H, new text p17, first paragraph).

Regarding ACC1 overexpression, we addressed the question of what role ACC1 plays in the tumorigenic phenotype of 4EBP1/2 deficient cells by performing a rescue experiment. In the initial manuscript, we knocked down ACC1 in 4EBP1/2 KO NT2197 cells and found increased tumorigenicity in vitro and in vivo as compared to control cells (Figure 5D, I&J). These data are in line with the Acc1 knockdown rescue experiments we performed using GL261 4ebp1 knockdown cells in an orthotopic xenograft model (Figure 6J). Together this highlights the importance that ACC1 inhibition plays in the pro-tumorigenic function of 4EBP1.

We understand that the reviewer is curious whether ACC1 overexpression will reduce tumorigenicity in vivo in this specific model. This question was addressed in a previous study using MCF7 cells (PMID: 22660331), where the authors showed that a constitutively active ACC1 reduced tumorigenicity in vivo.

Due to the large uncertainties linked to the situation in Israel, we are currently not able to perform in vivo experiments.

7. Figures 5 and 6: what are the glucose levels in tumors?

We appreciate the reviewer's question. Unfortunately, we did not measure glucose concentration in the tumors in Figures 5 and 6 as the samples were used for other experiments, including IHC and dityrosine immunoblots. Nevertheless, it is established that glucose is scarce

in tumor xenografts vs. normal tissues, as is exemplified here with tumor xenografts established in NOD *SCID* mice (PMID: 33596424).

REVIEWERS' COMMENTS

Reviewer #1 (Remarks to the Author):

The authors have largely addressed my comments.

Reviewer #2 (Remarks to the Author):

The authors have addressed my comments.

Reviewer #3 (Remarks to the Author):

The reviewers have addressed the concerns of this reviewer.

Reviewer #4 (Remarks to the Author):

The authors have addressed my previous comments. I have no additional concerns.